# Differential characterization of air ions in boreal forest of Finland and megacity of eastern China

Tinghan Zhang[1], Ximeng Qi[2], Janne Lampilahti[1], Liangduo Chen[3], Xuguang Chi[3], Wei Nie[3], Xin Huang[3], Zehao Zou[1], Wei Du[1], Tom Kokkonen[1,2], Tuukka Petäjä[1], Katrianne Lehtipalo[1,4], Veli-Matti Kerminen[1], Aijun Ding[2,3], and Markku Kulmala[1,2]

[1]Institute for Atmospheric and Earth System Research/Physics, Faculty of Science, University of Helsinki, Helsinki, Finland
[2]Najing-Helsinki Institute in atmospheric and Earth System Sciences, Nanjing University, Suzhou, China
[3]Joint International Research Laboratory of Atmospheric and Earth System Sciences, School of Atmospheric Sciences, Nanjing University, Nanjing, China
[4] Finnish Meteorological Institute, Helsinki, Finland

*Correspondence to*: Tinghan Zhang (tinghan.zhang@helsinki.fi) and Ximeng Qi (qiximeng@nju.edu.cn)

**Abstract.** Air ions play a crucial role in the new particle formation (NPF), which in turn has the potential to influence global climate and air quality. We conducted a comparative analysis of air ions in three size ranges (0.8-2 nm cluster ions, 2-7 nm intermediate ions and 7-20 nm large ions) at two "flagship" sites: the SMEAR II site in boreal forest of Finland and the SORPES site in megacity of eastern China. Air ion number size distributions (0.8–42 nm) were measured using a Neutral Cluster and Air Ion Spectrometer (NAIS) at these two sites from June 2019 to August 2020. At both sites, rising temperatures reduced the difference between positive and negative cluster ion concentrations, likely due to the enhanced convection and turbulent mixing that diminish the Earth's electrode effect. The median cluster ion concentration at SMEAR II (1270 cm$^{-3}$) was approximately six times higher than at SORPES (220 cm$^{-3}$), which was caused by the high coagulation sink in the urban area. The median large ion concentration at SORPES was nearly three times higher (197 cm$^{-3}$) than that at SMEAR II (67 cm$^{-3}$), which is due to the high number density of neutral aerosol particles facilitating ion attachment in the polluted megacity environment. The cluster ion concentration was negatively associated with the condensation sink (CS) at both sites, with a significantly stronger negative correlation at SORPES, suggesting that CS was a decisive factor for reducing the cluster ion concentrations in this urban area. The median formation rates of 2 nm ions at SMEAR II ($J_2^-$: 0.033 cm$^{-3}$ s$^{-1}$, $J_2^+$: 0.041 cm$^{-3}$ s$^{-1}$) were similar to those at SORPES ($J_2^-$: 0.028 cm$^{-3}$ s$^{-1}$, $J_2^+$: 0.025 cm$^{-3}$ s$^{-1}$). The median ion-induced fractions were 19.9% and 1.3% at SMEAR II and SORPES, respectively, indicating a minor contribution of ions to NPF in polluted environments. Nevertheless, the charged particles were activated earlier than neutral particles at SORPES, indicating that the ion-induced nucleation could precede neutral nucleation in this polluted environment. In addition, the contribution of ion-induced nucleation at SORPES was higher at low NPF intensity, implying the non-negligible roles of air ions in urban aerosol production. This study underscores the need for long-term observations of air ions in both pristine boreal forests and polluted urban environments.

# 1 Introduction

Air ions are electrically charged substances in the atmosphere, ranging from molecular clusters to large aerosol particles of varying chemical composition (Arnold et al., 1978). The air ions in the troposphere are formed mainly through ionization from cosmic radiation, radon decay, and gamma radiation (e.g. Bazilevskaya et al., 2008; Chen et al., 2016). The continuous existence of air ions has been noted by numerous observations, indicating their ubiquitous distribution throughout the troposphere (Eichkorn et al., 2002; Hirsikko et al., 2005b; Laakso et al., 2008; Dos Santos et al., 2015; Yin et al., 2023; Hirsikko et al., 2007c). Air ions have historically sparked interest in the field of atmospheric electricity, since their flow in the electric field of the Earth-atmosphere system provides the observable conduction current in the atmosphere (Wilson, 1921; Harrison and Carslaw, 2003; Israel, 1970). In the recent decades, interest in air ions among atmospheric aerosol community was fueled due to their role in new particle formation (NPF) (Yu and Turco, 2001; Kulmala and Kerminen, 2008; Wagner et al., 2017; Arnold, 2008; Kulmala et al., 2007; Kirkby et al., 2023; Stolzenburg et al., 2020).

NPF, starting with the formation of molecular clusters and their subsequent growth to larger sizes, can considerably contribute to the atmospheric aerosols loading (Kulmala et al., 2004; Kerminen et al., 2018; Kulmala et al., 2012), thus exerting a notable impact on air quality (Guo et al., 2014; Kulmala et al., 2022a) and climate (Boucher et al., 2013). Electric charge can increase stability and decrease evaporation rates of newly formed molecular clusters. The enhancement of NPF due to ions is known as ion-induced nucleation (Yu and Turco, 2001; Curtius et al., 2006; Laakso et al., 2002). Chamber studies have demonstrated that ion-induced nucleation is a key factor contributing to the total nucleation rate, under conditions of low precursor species concentrations and low temperatures (Kirkby et al., 2011; Riccobono et al., 2014; Kirkby et al., 2016). Wagner et al. (2017) observed that ions can enhance the nucleation process under atmospherically relevant conditions and chemical mixtures of precursors when the corresponding neutral clusters would not be stable. Lehtipalo et al. (2016) discovered that ions can increase the growth rates of sub-3 nm particles, thus facilitating the growth process of nanoparticles. Besides the chamber experiments, model results indicate that ion-mediated nucleation (which includes ion-induced nucleation but also takes into account interactions between ions and particles, e.g., ion–ion recombination) could be the dominant pathway for NPF (Yu and Turco, 2011; Yu, 2010). However, the contribution of ion-induced nucleation to NPF in ambient conditions remains unclear at present. At the rural sites in Hyytiälä, Hohenpeissenberg, and Melpitz, ion-induced nucleation was estimated to contribute about 10% to the total formation of 2 nm particles (Kulmala et al., 2010). In urban areas, such as Helsinki, Boulder, and Nanjing, the contribution of ion-induced nucleation NPF was estimated to be lower, in the range of 0.2-1.3% (Herrmann et al., 2014; Iida et al., 2006; Gagné et al., 2012). At high-altitude sites, such as Pallas and Jungfraujoch, ion-induced nucleation was estimated to contribute 20%–30% to NPF (Manninen et al., 2010), and in Antarctica, this contribution was estimated to reach approximately 30% (Asmi et al., 2010). In general, field observations suggest that the contribution of ion-induced nucleation to NPF varies from clean to polluted environments.

The overwhelming majority of air ions observations have been conducted in the pristine Nordic boreal forests, which are renowned sources of biogenic volatile organic compounds (BVOCs) (Mäki et al., 2019; Wang et al., 2018). Boreal forests are

relatively clean environments away from major anthropogenic pollution sources, offering an unmatched chance to explore atmospheric processes without the complications introduced by urban or industrial emissions. In the boreal forest of Hyytiälä in Finland, Laakso et al. (2004) discovered that charged nucleation can contribute to particle formation by favoring the growth of negatively charged clusters. Moreover, both the concentration and mean size of sub-3 nm ions in boreal forest exhibited nocturnal increases, which has been shown to be distinctly linked to oxidized organic molecules (Junninen et al., 2008b; Lehtipalo et al., 2011b; Rose et al., 2018; Huang et al., 2024). Intermediate ions have been found to serve as a strong indicator of NPF in a boreal forest environment, with high concentrations observed only during NPF events (Leino et al., 2016; Tammet et al., 2014). Compared to the several ion studies in the boreal forests, investigations of air ions in polluted urban areas remain sparse. The few studies conducted in urban areas have mainly focused on air ion concentrations, finding that ion concentrations are typically lower in polluted regions primarily because of larger ion sinks in these areas (Skromulis et al., 2017; Dos Santos et al., 2015; Ling et al., 2010b; Hirsikko et al., 2007c). However, Ling et al. (2010a) found that ion concentrations varied considerably inside the urban area, due to additional ion sources such as power lines. More limited research in urban environments have attempted to quantitively examine the role of ions in NPF (Yin et al., 2023; Iida et al., 2006; Herrmann et al., 2014).

The Yangtze River Delta (YRD) of eastern China is one of the world's largest clusters of adjacent megacities with a cumulative population of about 100 million people, providing a valuable opportunity for urban atmospheric research. Based on a four-month measurement campaign in Nanjing within YRD, Herrmann et al. (2014) suggested that the behavior of cluster ions might be associated with NPF processes in this urban area. The dense population and vehicle emissions in the YRD result in a high condensation sink, which can cause significant losses of ions to large particles in urban air (Yin et al., 2023; Dos Santos et al., 2015). Nevertheless, BVOCs emissions from the abundant broadleaf vegetation to the southern YRD, as well as expanding green urban areas, infuses biogenic emissions into this urban pollution hotspot (Liu et al., 2018; Wang et al., 2020). Therefore, the YRD, with its combination of anthropogenic and biogenic elements, presents a complex environment that hinders a comprehensive understanding of ions. The Pan-Eurasian Experiment (PEEX) science plan, released in 2015, designated the northern Eurasian Arctic-boreal region and China, particularly the megacities in Eastern China as "PEEX region", given its potential significant impact on global air quality and climate (Kulmala et al., 2015; Lappalainen et al., 2014). Within the framework of the PEEX plan, both the boreal forest of Finland and the YRD of eastern China are focal areas of interest.

The main goal of this study is to characterize the similarities and differences in ion characteristics and contribution of ion to NPF between pristine boreal forests and complex urban polluted environments. To this end, in situ data from two "flagship" stations in the PEEX region—SMEAR II (The Station for Measuring Ecosystem–Atmosphere Relations II) in the boreal forest of Finland and SORPES (The Station for Observing Regional Processes of the Earth System) in the YRD region of eastern China—were utilized for a comparative analysis. In this study, by comparing data from two sites over one year from June 2019 to August 2020, we aim to address the following questions: (a) How do ion size distributions and number concentrations of both negative and positive polarities behave at these two sites?, (b) Which factors influence air ion size distributions and

number concentrations at both sites?, and (c) What are the differences in ion characteristics (number concentration, formation rate, growth rate, ion-induced fraction) related to NPF between the two sites?

## 2 Methods

### 2.1 Measurement sites

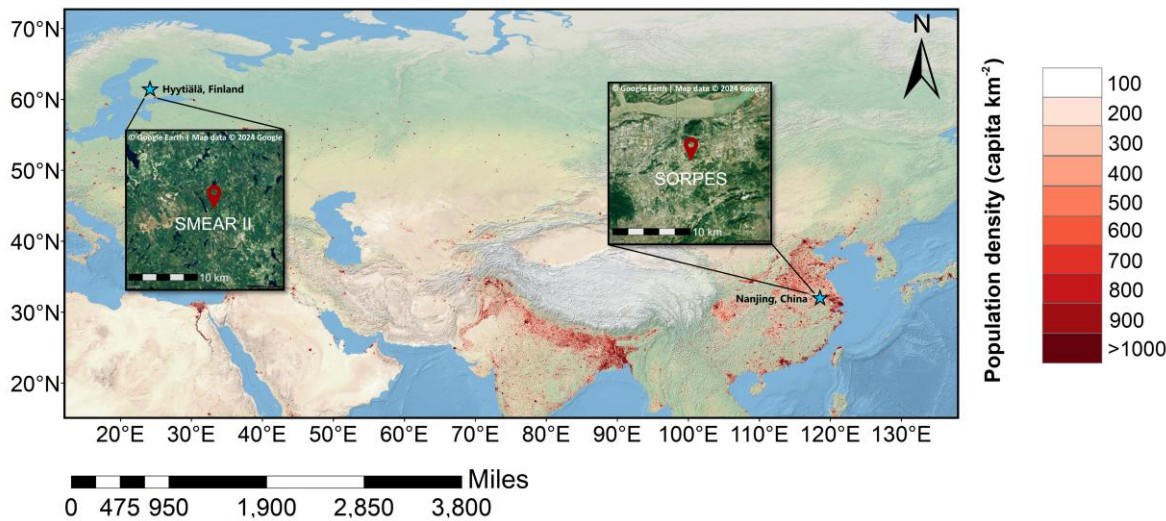

**Figure 1.** Locations of SMEAR II in Hyytiälä, Finland and SORPES in Nanjing, eastern China on the map (from Google Earth) with population density.

The air ions observed at the SMEAR II and SORPES stations were compared in this study (Fig. 1). SMEAR II is situated in a boreal Scots pine-dominated forest in Hyytiälä, southern Finland (61°51'N, 24°17'E; 181 m A.S.L. (Above Sea Level)). The station is located in a rural environment about 60 km northeast of Tampere, the nearest large city with a population of about 200,000. SMEAR II is renowned for conducting the world's longest continuous measurements of aerosol particle-number concentration and size distribution since 1996. Additionally, comprehensive observations of trace gases, soil-atmosphere fluxes, as well as meteorological variables, have been concurrently conducted at the site. More details about the station can be found from Hari et al. (2013).The SORPES station (32°07'N, 118°57'E; 40 m A.S.L.) is located in Nanjing, a megacity on the western edge of the Yangtze River Delta in eastern China. The station is situated in a suburban area about 20 km east of downtown Nanjing. Under the prevailing easterly wind throughout the year, the site tracks the background air in YRD region. Similar to the SMEAR II station, SORPES is equipped with comprehensive aerosol and meteorological instrumentation as well. A more detailed site description is presented by Ding et al. (2016)

## 2.2 Instrumentation

In this study, in situ data were collected from both SMEAR II and SORPES sites over a one-year-long time period, spanning from 7 June 2019 to 31 August 2020. At both sites, air ions and total particles data were measured at ground level using a NAIS (Neutral Cluster and Air Ion Spectrometer, decribed in Mirme and Mirme, 2013), which determines the number size distributions of ions and total particles in the electrical mobility diameter ranges 0.8-42 nm and 2.5-42 nm respectively.

The NAIS consists of two parallel DMAs (differential mobility analyzer), one for each polarity. Each polarity has a preconditioning unit containing an electrical filter and corona-needle charger in front of the DMA. In the ion mode, both positive and negative ions are simultaneously measured in the two columns. In the particle mode, aerosol particles are charged to opposite polarities by corona-needle chargers and then simultaneously measured in two DMAs. The measurement cycle was 2 min for both ion and particle mode, and 30 s for the offset mode. Along the body of each DMA, 21 insulated collector electrodes simultaneously detect and separate ions and charged particles according to their electrical mobility. The NAIS system utilizes a sample flow rate of 30 liters per minute (LPM) and a sheath air flow rate of 60 LPM for each analyzer. For particle data, measurements below about 2.5 nm are excluded from the measured size range due to contamination from charger ions. At both sites, each NAIS was placed indoors in an air-conditioned environment maintained at around 25 °C to ensure stable instrument temperatures and thus preventing potential temperature-induced biases or malfunctions. The NAIS data inversion was performed using the Spectops software following the procedure described by Manninen et al. (2016). Diffusion losses in the sampling tube were corrected after the data inversion. The NAIS data at SORPES from 14th September 2019 to 15th October 2019 was unavailable because the instrument was taken away for a short-term campaign.

At SMEAR II, particle number size distributions ranging from 3 to 1000 nm were additionally measured using a twin differential mobility particle sizer (DMPS). The twin DMPS measurement system contains two setups, each comprising a cylindrical DMA and a condensation particle counters (CPC). Detailed description for DMPS instrumentation can be found from Aalto et al. (2001). At SORPES, particle size distribution from 4 to 500 nm was measured by two Scanning Mobility Particle Sizers (SMPS, TSI Inc.), i.e. nano-SMPS and long-SMPS, which consisted of different DMA and CPC. Concurrently, an Aerodynamic Particle Sizer (APS, TSI Inc.) covered the size range from 500 nm to 20μm (aerodynamic diameter). The SMPS data were combined with the APS data according to the method described by Beddows et al. (2010) to obtain the particle number size distribution from 4 nm to 1000 nm used in this study. In addition to ion and aerosol data, this study also used meteorological data: air temperature, relative humidity, boundary layer height (BLH), wind speed, wind direction, and precipitation from two sites. A summary of the instrumentation utilized at two stations is provided in Table S1.

## 2.3 Data analysis

In this work, according to the protocol of the atmospheric electricity measurement community, air ions were mobility-classified as cluster or small ions (3.2-0.5 $cm^2$ $V^{-1}$ $s^{-1}$), intermediate (0.5-0.034 $cm^2$ $V^{-1}$ $s^{-1}$), and large ions (0.034-0.0042 $cm^2$ $V^{-1}$ $s^{-1}$), which correspond to mobility diameters of 0.8-2, 2-7, 7-20 nm, respectively. The traditional NPF event

classification followed the procedure presented by Dal Maso et al. (2005) using the DMPS data. Concisely, we categorized the days into 4 types: NPF event I days (the days exhibiting a discernible "banana" shaped curve in the particle size distribution temporal surface plot, allowing for the calculation of particle growth and formation rates), NPF event II days (the days indicative of NPF activity, yet lacking precise calculability of growth and formation rates), undefined days (the days where the occurrence of NPF is ambiguous), and non-event days (the days with no evidence of NPF events). Concurrently, the number concentrations of 2.5-5 nm particles observed by NAIS at each site were employed in newly developed Nano Ranking Analysis. This novel method group days into a number percentile intervals to probabilistically characterize the occurrence and intensity of NPF at the two sites. The procedures and detailed description of the "Nano Ranking Analysis" method were presented in Aliaga et al. (2023).

The condensation sink (CS) accounts for the loss rate of vapor molecules due to the condensing onto existing aerosol particles in the atmosphere (Kulmala et al., 2012).The CS can be expressed as:

$$CS = 2\pi D \sum_{d_p} \beta_{m,d_p} d_p N_{d_p}, \tag{1}$$

where $D$ is the diffusion coefficient of the condensing vapor, $\beta_{m,d_p}$ is transition−regime correction, $d_p$ and $N_{d_p}$ are the geometric mean diameter of particles and particle number concentration in each size bin, respectively.

The calculation of particle growth rates and formation rates follow the procedure described in Kulmala et al. (2012). The growth rate (GR) of a particle population during the NPF events can be expressed as:

$$GR = \frac{dd_p}{dt} = \frac{\Delta d_p}{\Delta t} = \frac{d_{p2} - d_{p1}}{t_2 - t_1}, \tag{2}$$

where the $d_{p1}$ and $d_{p2}$ represent the particle diameters in the unit of nm at the time $t_1$ and $t_2$, respectively. For calculation, $d_{p1}$ and $d_{p2}$ refer to the center of the size bin and $t_1$ and $t_2$ are the times the concentration of this size bin reaches the maximum. The GR of ions and particles in the size ranges of 3-7, 7-20 nm were calculated using the NAIS data.

The particle formation rate $J_{d_p}$ of diameter $d_p$ is calculated according to the equation given by Kulmala et al. (2012):

$$J_{d_p} = \frac{dN_{d_p}}{dt} + CoagS_{d_p} \times N_{d_p} + \frac{GR}{\Delta d_p} \times N_{d_p}, \tag{3}$$

where the first term on the right−hand side is the time evolution of the particle number concentration in the size ranging from $d_p$ to $d_p + \Delta d_p$. The second term is the coagulation sink ($CoagS_{d_p}$) for particles between $d_p$ to $d_p + \Delta d_p$. The coagulation sink is the loss rate of ions/particles due to their coagulation with larger particles. The third term represents the growth out of the corresponding size range where GR is the respective growth rate. For the calculation of the formation rate of total 2.5 nm particles and 2 nm ions, the GR of particles and ions in the size range of 3-7 nm were used.

Two additional terms need to be considered when calculating the formation rate of negatively (superscript −) and positively (superscript +) charged particles, $J_{d_p}^{\pm}$, according to the equation below (Kulmala et al., 2012):

$$J_{d_p}^{\pm} = \frac{dN_{d_p}^{\pm}}{dp} + CoagS_{d_p}N_{d_p}^{\pm} + \frac{GR}{\Delta d_p}N_{d_p}^{\pm} + \alpha N_{d_p}^{\pm}N_{<d_p}^{\mp} - \beta N_{d_p}N_{<d_p}^{\pm}, \tag{4}$$

The additional terms to be included account for the loss due to the ion-ion recombination (fourth term on the right side) and gain via the attachment of neutral particles to smaller ions (fifth term on the right side). The ion-ion recombination coefficient $\alpha$ and the ion-aerosol attachment coefficient $\beta$ can be derived from either theory or measurement, and in principle they depend on the shape of particle size distribution. In this work, the rate coefficients $\alpha$ and $\beta$ were assumed based on the typical values $1.6 \times 10^{-6}$ cm$^3$ s$^{-1}$ and $0.01 \times 10^{-6}$ cm$^3$ s$^{-1}$, respectively (Franchin et al., 2015; Tammet and Kulmala, 2005).

The ion-induced fraction of NPF at 2 nm, which indicates the contribution of ion-induced nucleation to the overall nucleation rate, was calculated with a modified equation based on the earlier work by Manninen et al. (2010):

$$\text{Ion-induced fraction [2 nm]} = \frac{J_2^+ + J_2^-}{J_{2.5}^{total}}, \tag{5}$$

where $J_{2.5}^{total}$ and $J_2^{\pm}$ are calculated using the Eqs. (3) and (4). In previous studies, the total particle formation rate was typically taken at 2 nm in previous studies. To avoid contamination from charger ions in the NAIS total particle measurements, the total particle formation rate was calculated at 2.5 nm in this study. This methodological difference might introduce a slight overestimation, since the formation rate at 2.5 nm is theoretically lower than that at 2 nm. To maintain consistency with previous measurements in the boundary layer (Gagné et al., 2008; Manninen et al., 2010; Kulmala et al., 2013), our analysis of ion-induced fraction does not consider particles formed by ion-ion recombination. Consequently, this work considered solely the particles that were charged at 2 nm when assessing the ion-induced fraction.

## 3 Results and discussion

### 3.1 Characteristics of ion concentration at SMEAR II and SORPES

#### 3.1.1 Ion number size distribution and polarity at two sites

As shown in Fig. 2a-b, the median ion number size distributions (INSDs) for both negative and positive polarity within the 0.8-40 nm diameter range exhibited comparable patterns. A remarkable peak occurred in the 0.8-2 nm range, corresponding to cluster ions. The prominence of this size range is attributed to the continuous formation of cluster ions primarily through the ionization of neutral molecules and clusters and particles throughout the troposphere (Hirsikko et al., 2011). The peak of cluster ions was followed by a dramatic decline, even approaching zero, in the 2-7 nm intermediate ion size range, and then an uptick in the 8-40 nm large ion size range. The intermediate ion concentration is typically very low, being detectable only during NPF events and under specific conditions such as snowfall and rainfall (Hirsikko et al., 2011; Tammet et al., 2009; Laakso et al., 2007). In the atmosphere, cluster ions tend to rapidly attach to aerosol particles, facilitating the formation of larger ions, which can explain the elevated INSDs in the large ion size range. At SORPES, the peak INSD in the cluster ion size range was significantly lower than that at SMEAR II within the same size range. This difference is because cluster ions are removed faster by coagulation with high concentration of pre-existing aerosol particles in a polluted environment compared

with a clean environment, thus resulting in lower cluster ion concentrations at SORPES. Additionally, the abundance of cluster ions at SMEAR II may be due to the higher ion production rate via higher emissions from radon decay or stronger external radiation. Conversely, the peak INSD at SORPES in the large ion size range was considerably higher than that observed at SMEAR II. Since cluster ions are continually produced in the atmosphere and have a high coagulation probability with bigger
neutral particles in the air, this attachment process may serve as a substantial source of large ions. At polluted SORPES, aerosol particles from heavy traffic could further facilitate this process, resulting in higher concentrations of larger ions. Previous studies have also observed that large ions are associated with traffic emissions in urban environments (Dos Santos et al., 2015; Hirsikko et al., 2007c; Tiitta et al., 2007; Thomas et al., 2024).

Significant differences between the negative and positive INSDs were observed in the cluster ion size range at both sites
(Fig. 2c-2d). At SMEAR II and SORPES, the peaks of INSDs in the cluster size range for positive ions were shifted slightly to larger sizes compared to negative ions (Fig. 2a-2b), indicating that the mean size of positive cluster ions was larger than that of negative cluster ions. At SMEAR II, the concentrations of positive and negative cluster ions were comparable, consistent with previous observations in Hyytiälä (Sulo et al., 2022; Komppula et al., 2007; Hirsikko et al., 2005a) using balanced scanning mobility analyzer (BSMA, Tammet, 2006) and air ion spectrometer (AIS, Mirme et al., 2007). In contrast, at
SORPES, the concentration of positive cluster ions was higher than that of negative cluster ions. This phenomenon may be attributable to SORPES hosting more air ions with the compounds containing the highest proton affinities, allowing them to capture positive charges. The higher concentration of positive cluster ions at SORPES may also be partly due to the electrode effect of the negatively charged Earth's surface, which typically attracts small positive ions downward, causing a dominance of positive polarity near the ground in calm air (Wilson, 1924; Hoppel, 1967). Air temperature was found to be a crucial factor
affecting ion polarity at SMEAR II and SORPES. At both sites, as air temperature increased, the difference between the concentrations of negative and positive cluster ions decreased. This phenomenon might be due to increased convective motions and turbulent mixing at higher air temperatures, which reduces the Earth's electrode effect.

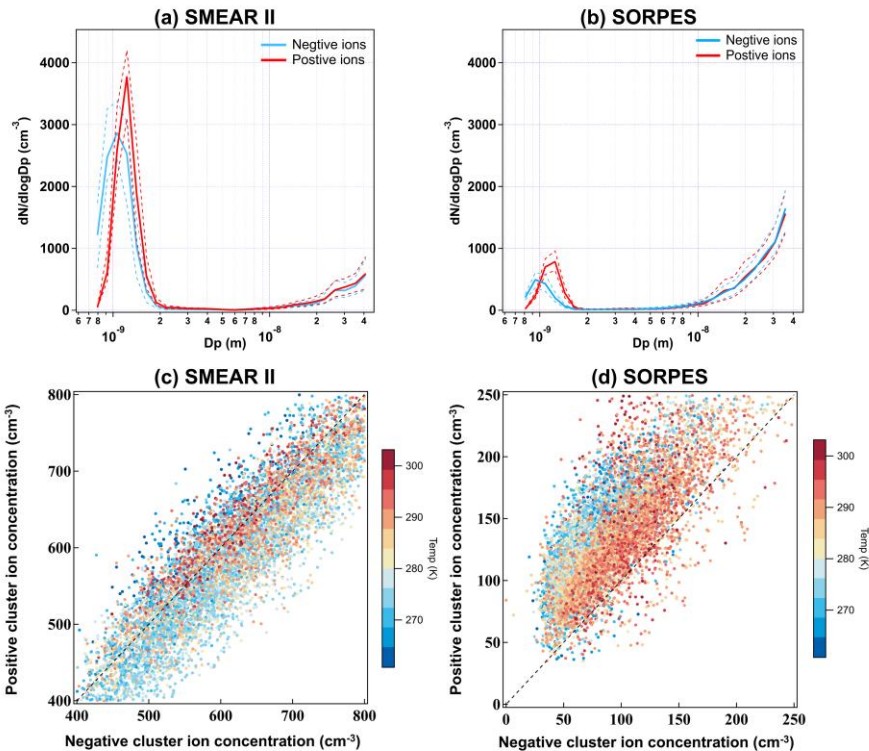

**Figure 2.** The median ion number size distribution of negative and positive polarity at (a) SMEAR II and (b) SORPES, the solid line indicates the median, and the dash lines indicate 25th and 75th percentile distributions. The correlation between median positive and negative cluster ions at (c) SMEAR II and (d) SORPES, the dots are colored by air temperature. The black dash line is 1:1 line.

### 3.1.2 Seasonal and diurnal variations of ion concentration at two sites

Figure 3 shows clear seasonal variations in the three size ranges (cluster: 0.8-2 nm, Intermediate: 2-7 nm, Large: 7-20 nm) at the two sites. At both sites, although positive and negative cluster ion concentrations differed from each other, both polarities shared similar seasonal patterns. Notably, a significant dissimilarity in cluster ion concentrations were observed between the two sites (Fig. 3a-3b). Throughout the whole study period, the median concentration of total cluster ions (sum of both polarities) was 1270 cm⁻³ at SMEAR II and 220 cm⁻³ at SORPES. These values fall within the range of 200-2500 cm⁻³ reported in a review by Hirsikko et al. (2011) for sites all over the world. The total cluster ion concentration at SORPES based on our observation is notably lower than that reported by Herrmann et al. (2014), which ranged between 600 and 1000 cm⁻³. In this study, the median CS (0.020 s⁻¹) was found to be lower compared to the value presented in Herrmann et al. (2014) (0.041 s⁻¹), which theoretically should lead to an increase in cluster ion concentrations. However, the observed decrease suggests that the sources of cluster ions may have changed. A possible explanation could be the rapid urbanization in Nanjing

over the past decade, including the transformation of unpaved roads to cemented surfaces, may have led to a reduction in radon emissions from the ground. Although the intercomparison study between the AIS and NAIS showed reasonably good agreement in cluster ion concentrations for outdoor air (Gagné et al., 2011), it is possible that the difference is partly due to instrument bias, as Herrmann et al. (2014) conducted ion measurements with an AIS. At both sites, cluster ion concentrations generally showed an increase during the summer months, with the highest concentrations for both polarities measured in August at SMEAR II and in October at SORPES (Fig. 3a-3b). Given that the near-ground cluster ions primarily originate from cosmic rays, radon and gamma radiation from soil (e.g. Hirsikko et al., 2011), the seasonality of cluster ions is likely linked to the seasonal change in radon exhalation, which is influenced by factors such as snow depth, soil humidity, and boundary layer height (Lopez et al., 2012; Shashikumar et al., 2008; Hatakka et al., 1998). Additionally, the evolution of the boundary layer may affect the distribution and availability of precursor vapors, potentially influencing variations in cluster ion concentration and size.

The intermediate ion concentrations were relatively similar and very low at the two sites (Fig. 3c-3d). During the experiment period, the median total intermediate ion concentration at SMEAR II and SORPES were 27 cm$^{-3}$ and 25 cm$^{-3}$, respectively. The intermediate ion concentrations increased significantly in the spring at SMEAR II and in the autumn at SORPES. Since intermediate ions are predominantly detected on NPF event days, and NPF events were found to mainly occur during spring and autumn at each sites in both this study(see section 3.3.1) and previous studies (Nieminen et al., 2014; Qi et al., 2015), the observed higher intermediate ion concentration at the two sites during these periods were expected. An intriguing behavior in the seasonal variation of intermediate ions at SORPES can be seen in Fig. 3d, with the negative ion concentrations significantly surpassing those of positive ions in June and July. As rain and waterfalls have been found to produce negatively charged particles smaller than 10 nm (Tammet et al., 2009), the high-level of ambient negative ion at SORPES in summer can be attributed to heavy and intensive rainfall in YRD of China (Fig. S1). In addition, throughout the entire measurement period, negative intermediate ion concentrations on rainy days were consistently higher than those during non-rainy periods at both sites (Fig. S2).

The median total concentration of large ions was 67 cm$^{-3}$ at SMEAR II and 197 cm$^{-3}$ at SORPES. The higher concentration of large ions at SORPES can be ascribed to heavy traffic emissions in urban areas, acting as a source for large ions. This finding aligns with studies near busy roads by Hirsikko et al. (2007c) and Tiitta et al. (2007), which suggest that large ion concentrations are affected by traffic-related aerosols. Besides, the high levels of background aerosol loading at SORPES are likely conducive to the formation of larger particles via small ions attaching to larger ones, thereby contributing to an increased concentration of large ions. The large ion concentrations at SMEAR II peaked in spring, whereas the large ion concentrations at SORPES had multiple peaks in the seasonal variation. This is probably due to the complicated sources of aerosols, including the primary emissions and NPF in polluted environment (Fig. 3e-3f). A noticeable decrease in the large ion concentration at SORPES in February was observed (Fig. 3f), possibly caused by the substantial reduction in primary and vehicle emissions during the Chinese New Year period. The concentration of negative large ions at SORPES showed the same pattern as

intermediate negative ions in June and July, with concentrations clearly exceeding those of positive ions (Fig. 3f). This may be due to negative intermediate ions produced by rain attaching to background particles and growing in size.

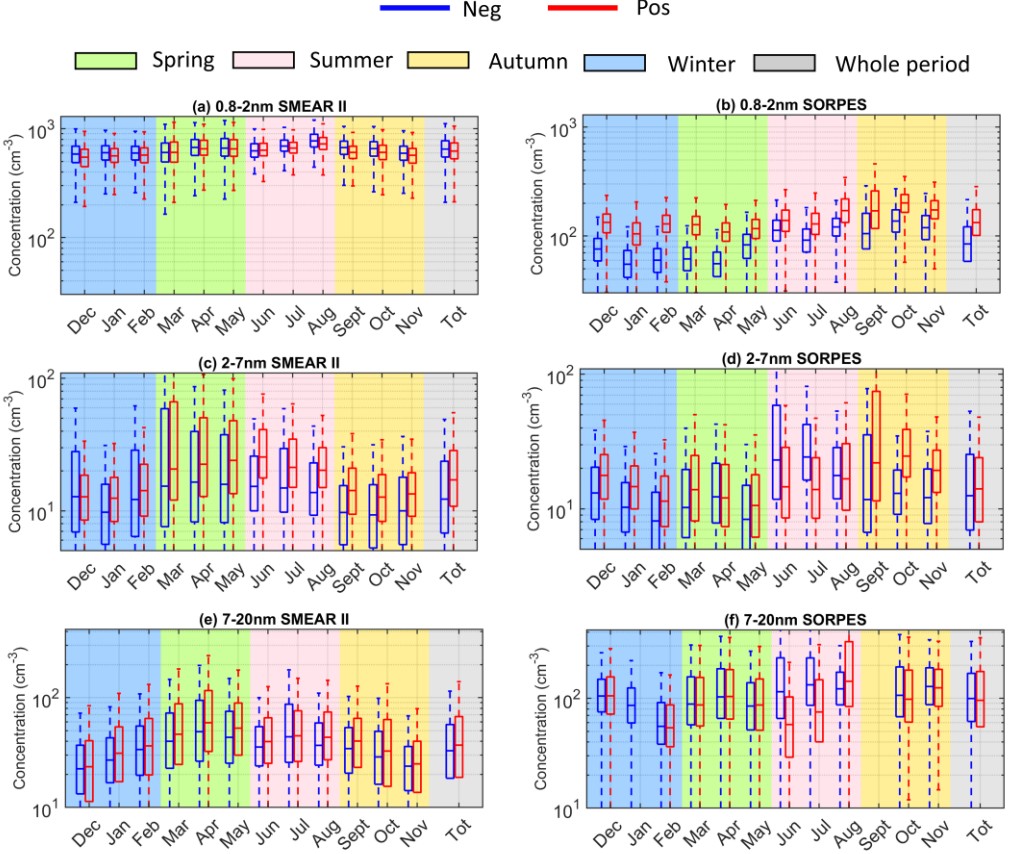

**Figure 3.** Median seasonal variations of negative and positive ion concentration in three size classes (cluster: 0.8-2 nm; intermediate: 2-7 nm; large: 7-20 nm) at SMEAR II (a, c, e) and SORPES (b, d, f). The line inside the box is the median; the top and bottom of each box are the 25th and 75th percentiles, respectively; the whiskers are 1.5 interquartile range. Negative and positive ion concentrations are depicted by the blue and red color, respectively. Different background colors in the figure represent various seasons, with grey indicating data for the whole period. Positive large ion concentration in January, September, and negative large ion concentration in September at SOPRES were removed due to the electrometer contamination in corresponding size range (Fig. 3f). Note that NAIS data were unavailable from 14 September 2019 to 15 October 2019 at SORPES due to instrument deployment in a short-term campaign.

      Figure 4 illustrates the monthly diurnal variations in total ion concentration of different size classes at SMEAR II and SORPES. At both sites, the cluster ion concentration remained consistently low during winter and barely showed clear diurnal

variations (Fig. 4a-4b). At SMEAR II, discernible diurnal cycles of cluster ion concentration were observed during the warm months (April to September), characterized by a significant increase during the night between 20:00 and 4:00 LT (Fig. 4a). This nocturnal rise in cluster ion concentration agrees with study by Mazon et al. (2016) based on the 11-year ion measurement at Hyytiälä, with a night-time formation of 0.9-2.4 nm ions that can surpass corresponding ion concentration levels during daytime. Chen et al. (2016) suggest that the nighttime build-up of cluster ions at SMEAR II may be due to the enhanced charge acquisition. Additionally, this increase could be linked to the accumulation of ionizing radiation from radon decay, attributed to lower boundary layer mixing heights before sunrise (Hirsikko et al., 2011). Apart from elevated production, such nighttime increase may also result from a weakening of the sinks for these ions, indicating that the removal processes of cluster ions are suppressed. However, these processes are modulated by prevailing atmospheric conditions. Therefore, the phenomenon presented in Fig. 4a may be the result of the synergy between the production and consumption mechanisms of cluster ions and atmospheric dynamics. At SORPES, despite a general increase in cluster ion concentration during warmer months compared to colder seasons, no distinct diurnal cycles were observed (Fig. 4b). The absence of nighttime, and even daytime, bursts in cluster ion concentration at SORPES may be attributed to high background particle loading, which likely keeps the sink of cluster ions consistently high throughout the day.

At both sites, diurnal variations in intermediate ion concentrations were only prominent during the warm seasons (Fig. 4b-4c), characterized by a pronounced peak around noon (11:00-13:00 LT). This is because the variations in intermediate ion concentrations are closely linked to NPF, which typically occurs during the period of the strongest photochemical oxidation around midday. In addition, at SMEAR II, intermediate ion concentrations increased during the night, coinciding with the nighttime increase of cluster ions (Fig. 4a). This suggests that such increase in intermediate ion concentration was likely due to the growth of small ions or the attachment of cluster ions to growing neutral particles associated with NPF. In contrast to cluster ions, the nighttime increase in intermediate ions primarily occurred from March to May, which may be attributed to abundant biogenic emissions from the boreal forest during spring. Some studies have shown that the nocturnal bursts of intermediate ions during spring months in Hyytiälä correlate well with the concentration and oxidation products of monoterpenes (Eerdekens et al., 2009; Lehtipalo et al., 2011a; Rose et al., 2018; Huang et al., 2024). At both sites, the diurnal patterns of large ion concentrations were similar to those of intermediate ions, with a peak during the day in warmer seasons and a noticeable increase at night at SMEAR II (Fig. 4e-4f). This pattern suggests that large ions at two sites may originate from not only the coagulation of cluster ions to larger particles, but also the growth of intermediate ions.

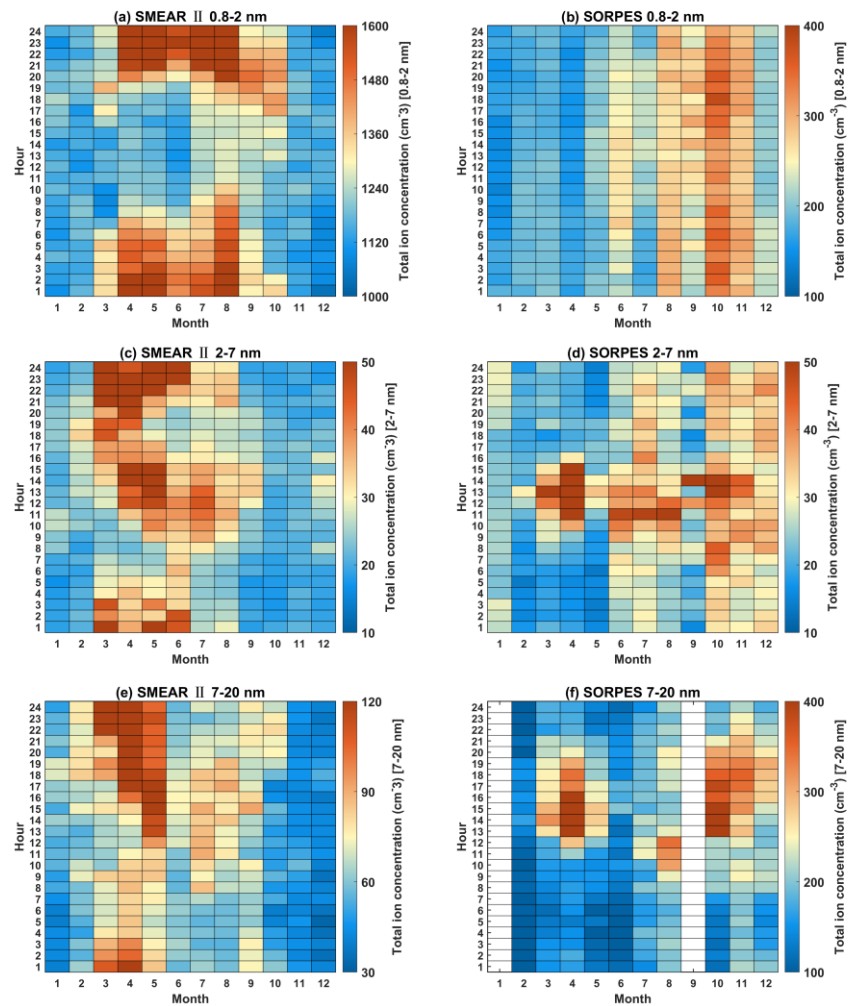

**Figure 4.** Monthly median diurnal variations of total ion concentration (sum of both polarity) of three size classes (cluster: 0.8-2 nm; intermediate: 2-7 nm; large: 7-20 nm) at SMEAR II (a,c,e) and SORPES station (b,d,f) from June 2019 to August 2020. large ion concentration in January and September at SOPRES were removed due to the electrometer contamination in corresponding size range (as shown in Fig. 3f). Note that the color scales are different for two sites. NAIS data were unavailable 330 from 14 September 2019 to 15 October 2019 at SORPES due to instrument deployment in a short-term campaign.

### 3.2 Factors influencing ion concentration

We investigated the correlations of meteorological factors (air temperature, relative humidity, boundary layer height, wind speed) and CS with ion concentrations of both polarities across the three size ranges (As shown in Fig. S3). CS, as the widely recognized sink of ions to larger particles, showed a negative correlation with both positive and negative cluster ion 335 concentrations at both sites (Fig.S3). Concurrently, a negative association between total cluster ion concentration and CS at

two sites was seen (Fig. 5). These results indicate that CS is an important factor affecting the concentration of cluster ion. The median values of CS at SMEAR II and SORPES were 0.0025 s$^{-1}$ and 0.0197 s$^{-1}$, respectively, with the CS at polluted SORPES being nearly eight times higher than that at the clean SMEAR II. This highlights the significantly higher aerosol loadings in urban areas, leading to greater coagulation losses of cluster ions to larger particles. This substantial difference in CS is likely

the main reason for the significantly lower cluster ion concentration at SORPES compared to SMEAR II, as discussed in previous sections. Notably, the negative association was more pronounced in the urban (SORPES) than in the clean (SMEAR II) environment. Also, the negative correlation coefficient between cluster ion concentration and CS was considerably higher at SORPES compared to SMEAR II (Fig. S3). At SMEAR II, the lack of a strong CS dependency of cluster ions could be due to the low background aerosol loading in the clean forest environment. Studies at SMEAR II found that changes in cluster ion

concentrations at this clean site are primarily determined by changes in the ion production rate, which in turn is mainly driven by ionizing radiation and weather conditions, thus rendering CS a less important factor in cluster ion concentration (Chen et al., 2016; Hirsikko et al., 2007a). Although Sulo et al. (2022) found that CS might explain the long-term trend of cluster ion concentrations in boreal forest environment, short-term variations were more complex to explain, and the dependence of cluster ions on the CS varied across different seasons. In line with our study, Yin et al. (2023) identified a stronger negative correlation

between CS and cluster ion concentration in Beijing. These results indicate that in clean areas with large variations in ion production rates and low aerosol mass loadings, the ion production rate may be the controlling factor for cluster ion concentration. However, in urban areas with high background particle loading, CS might be the decisive factor affecting cluster ion concentrations.

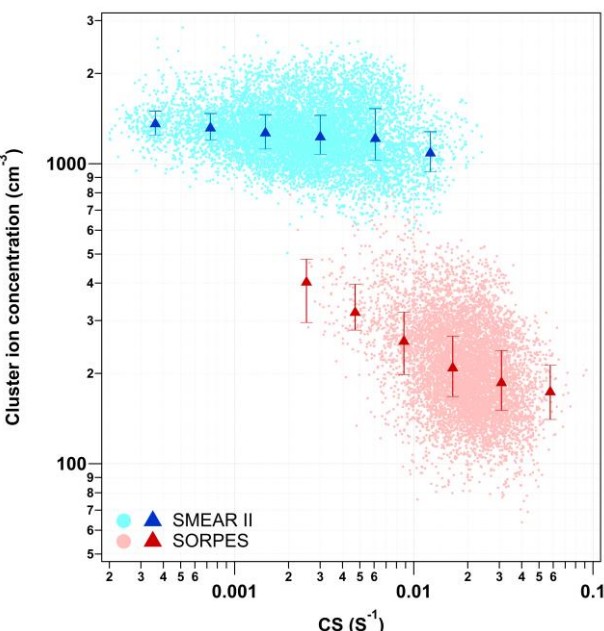

**Figure 5.** Scatter plots of the logarithm of total cluster ion concentration (sum of the polarity) versus the logarithm of condensation sink (CS) at SMEAR II (blue dots) and SORPES (red dots). Blue and red markers and error bars represent the median values and 25%-75% ranges in each CS bin for SMEAR II and SORPES, respectively.

Among the meteorological factors investigated in our study, air temperature showed the strongest positive correlation with cluster ion concentrations, and this correlation was stronger for negative than positive ions at both SMEAR II and SORPES (Fig. S3). This supports the earlier finding that the difference between the positive and negative ion concentrations decreases as temperature increases. In addition, wind speed showed a significant negative correlation with cluster ion concentrations at SMEAR II. By examining the interrelationships between the wind speed, wind direction and ion concentrations across the three size ranges, we further demonstrated the impact of wind on ion concentrations at both sites (Fig. 6).

At SMEAR II, cluster ion concentrations decreased significantly with an increasing wind speed, but increased when the wind was blowing from the northwest (Fig. 6a). Both intermediate and large ion concentrations were higher during northwesterly and southwesterly winds at SMEAR II, but this increase was only observed at higher wind speeds (Fig. 6c and 6e). Wind directions from the northerly sector favour NPF events at this site, as air masses from the clean marine areas to the north have a lower CS (Nieminen et al., 2014; Nilsson et al., 2001). Therefore, NPF could be a contributing factor to the elevated concentrations of air ions observed during northwesterly winds. The peak in intermediate ion concentrations observed in the southwest may be associated with frequent precipitation from that direction, in accordance with the findings which have demonstrated the frequency of rain-induced intermediate ion bursts in the boreal forests (Hirsikko et al., 2007b) (Fig. 6c). At SORPES, cluster ion concentration was higher during northerly and southwesterly winds (Fig. 6b). Additionally, intermediate and large ion concentrations were significantly higher during westerly winds compared to easterly winds, exhibiting a remarkable rise in the southwest wind direction (Fig. 6d and 6f). As mentioned in section 2.1.2, the SORPES site is situated southwest of main traffic roads. Chen et al. (2023) observed that the highest concentrations of $NO_x$ were recorded in that air mass direction at SORPES, indicating that traffic emissions may constitute a major source of larger ions. Roadside studies conducted in Finland also revealed that traffic emission can increase intermediate and large ion concentrations (Hirsikko et al., 2007c; Tiitta et al., 2007). Consequently, in polluted urban areas, in addition to NPF, traffic-produced aerosols could be an important factor affecting ion concentrations.

Nevertheless, no single meteorological factor in our study showed a particularly strong correlation with ion concentrations (Fig. S3). This is expected, as ion concentrations are affected by the interplay between ion sources, sinks, and meteorological conditions. Therefore, to fully understand the reasons behind variations in ion concentration in different environments, simultaneous observation and analysis of these factors are essential.

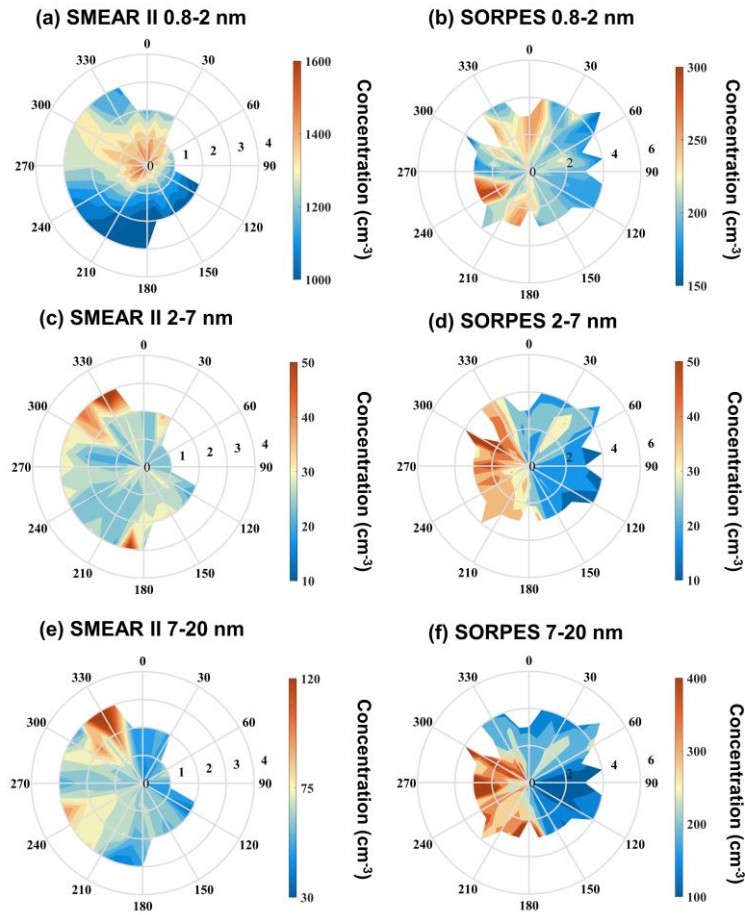

**Figure 6.** Total ion concentration (sum of both polarity) in three classes (cluster: 0.8-2 nm; intermediate: 2-7 nm; large: 7-20 nm) as a function of wind direction and wind speed at SMEAR II (a,c,e) and SORPES (b,d,f).

### 3.3 Ions and new particle formation

### 3.3.1 NPF at SMEAR II and SORPES

The overall frequency of NPF events (sum of event I and event II) at SMEAR II was 16%, with the highest occurrence in spring with the value of 43% (Fig. 7a). This frequency is somewhat close to the 1996-2012 period reported by Nieminen et al. (2014)(i.e., 23%), which similarly noted that NPF is most frequent during the spring months in Hyytiälä. Even though the high concentration of pre-exiting particles in urban environment was expected to inhibit NPF, NPF occurred even more frequently at SORPES. The overall frequency of NPF events at SORPES during the entire measurement period was 39%, with higher frequencies in the warm seasons (spring, summer and autumn) (Fig. 7b). The overall NPF frequency at SORPES during the measurement period from 2019 to 2020 is in close agreement with the 44% and 41% frequencies reported by Qi et al. (2015) during 2012-2013 and Chen et al. (2023) during 2018-2020, respectively.

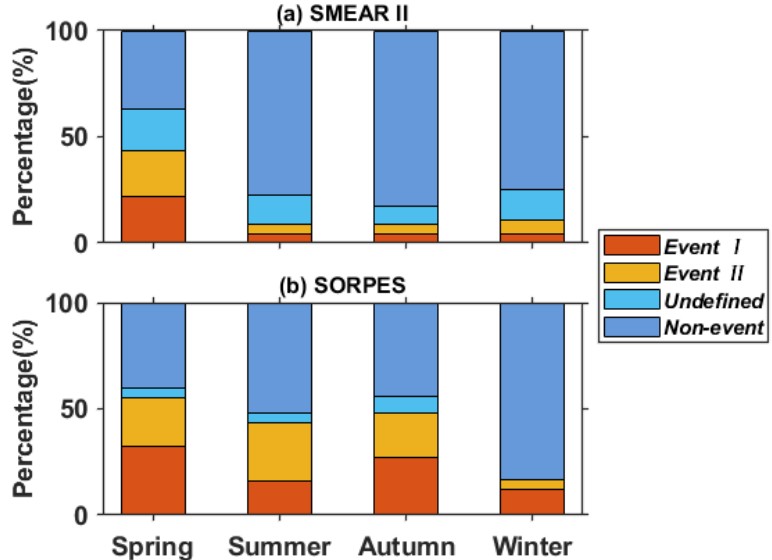

**Figure 7.** Seasonality of NPF frequency from June 2019 to August 2020 at SMEAR II (a) and SORPES (b).

To further investigate the relationship between ions and NPF, the diurnal variations of median total ion concentrations in the three size classes between NPF event days and non-event days at SMEAR II and SORPES were compared (Fig. 8). Generally, at both sites, distinct diurnal patterns in ion concentrations were observed during the NPF event days: a peak in the cluster ion concentration at SMEAR II occurred at night (22:00 LT), while at SORPES the highest concentration was observed in the early morning (7:00 LT), several hours before typical time of NPF (9:00-12:00 LT) (Fig. 8a-8b). Additionally, at both

sites, cluster ion concentrations prominently decreased during the NPF event days in the afternoon, even falling below non-event day levels. At both sites, intermediate ion concentrations during the NPF event days showed significant increases, reaching values 8-14 times higher than those at the same time on the non-event days (Fig. 8c-d). This substantial increase not only agrees with Leino et al. (2016) in that intermediate ions can serve as an effective indicator for NPF at clean SMEAR II, but also highlights their sensitivity to NPF in polluted areas. The peak in intermediate ion concentration at SORPES (11-12:00

LT) appeared earlier than at SMEAR II (13:00 LT). At SMEAR II, intermediate ion concentrations increased considerably during the nights following NPF (18:00-24:00 LT), which might be related to nocturnal ion clustering in the boreal forest. Such nocturnal increases have been observed to be more likely following a NPF event day than a non-event day (Junninen et al., 2008a). However, nocturnal ions clustering was not observed at SORPES on either NPF event or non-event days. For large ions, distinct increases were also noted during NPF event days at both sites, with the peak appearing earlier at SORPES (13:

00 LT) than at SMEAR II (18:00 LT) as well (Fig. 8e-8f). This timing difference could be partly caused by a higher growth rate of newly formed particles at SORPES (see Section 3.3.2). In addition, the rapid increase in larger ion concentration may be attributed to the slower growth of boundary layer height on NPF event days at SORPES, which facilitates the augmentation in ion concentration (Fig. S4). As shown in Fig. 8f, slight increases in large ion concentration were observed at SORPES during

the morning (08:00-10:00 LT) and afternoon (15:00-18:00 LT) on non-event days, potentially originating from traffic emissions during the morning and evening rush hours near the site.

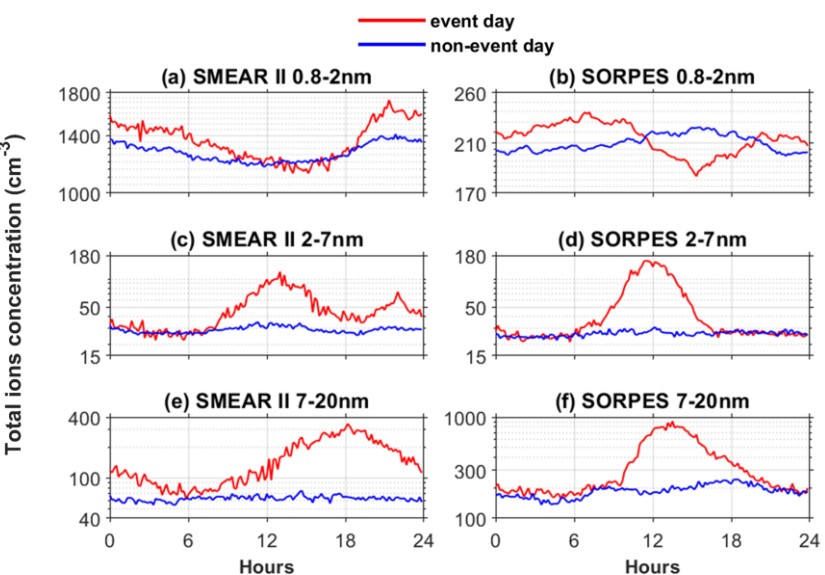

**Figure 8**: Median diurnal cycles for total ion concentrations at SMEAR II and SORPES in three size ranges, separated by NPF event and non-event days. (a-b) Cluster ions: 0.8-2 nm. (c-d) Intermediate ions: 2-7 nm (e-f) Large ions: 7-20 nm.

### 3.3.2 Ion formation rate and growth rate

We determined the formation rates of 2 nm and 3 nm ions, as well as the growth rates (GR) for ions from 3 to 7 nm and from 7 to 20 nm at SMEAR II and SORPES (Table 1). According to our best knowledge, this work represents the first long-term formation rate and growth rate of charged particles in the western part of the YRD region. The median values of $J_2^{\mp}$ and $J_3^{\mp}$ for both polarities during the active time (9:00-15:00 LT) at each NPF event were not significantly different between SORPES ($J_2^-$: 0.028 cm$^{-3}$ s$^{-1}$, $J_2^+$: 0.025 cm$^{-3}$ s$^{-1}$; $J_3^-$: 0.028 cm$^{-3}$ s$^{-1}$, $J_3^+$: 0.027 cm$^{-3}$ s$^{-1}$) and SMEAR II ($J_2^-$: 0.033 cm$^{-3}$ s$^{-1}$, $J_2^+$: 0.041 cm$^{-3}$ s$^{-1}$; $J_3^-$: 0.012 cm$^{-3}$ s$^{-1}$, $J_3^+$: 0.016 cm$^{-3}$ s$^{-1}$). This finding is consistent with previous results obtained across 12 European sites (Manninen et al., 2010), showing that the charged formation rate at 2 nm varies little between different sites, typically ranging between $10^{-2}$ and $10^{-1}$. In contrast, the total particle formation rate at 2 nm varies considerably across sites, even by more than an order of magnitude. It indicates that neutral particle formation rates are more sensitive to surrounding environmental conditions than ion formation rates. Figure 9 displays the clear seasonal variations in 2 nm and 3 nm ion formation rates of both polarities during the entire measurement period at SMEAR II and SORPES. At both sites, $J_2^{\mp}$ and $J_3^{\mp}$ exhibit similar seasonal patterns, with higher values observed during the warmer seasons. The formation rate of 2 nm and 3 nm ions peaked in the spring at SMEAR II and in the summer at SORPES. The higher ion formation rates during the warmer part of the year may be associated with increased biogenic emissions and stronger atmospheric oxidation capacity.

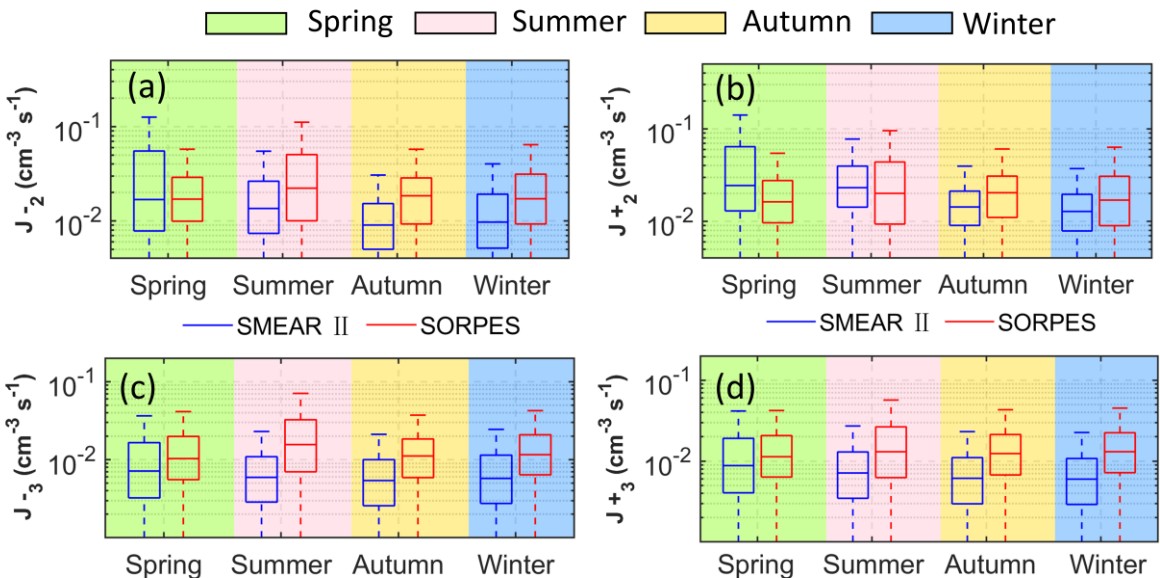

**Figure 9**: Seasonal variation of the formation rate of 2 nm ($J_2^\mp$, a-b) and 3 nm ions ($J_3^\mp$, c-d) at SMEAR II and SORPES sites for both negative and positive polarities during the entire measurement period. The line inside each box is the median; the top and bottom of each box are the 25th and 75th percentiles, respectively; the whiskers are 1.5 interquartile range.

The GR of ions were size-dependent at SMEAR II and SORPES, with a clear increase with an increasing size of the ion. Similar results were found from K-puszta (Yli-Juuti et al., 2009) in Hungary, Tumbarumba in Australia (Suni et al., 2008) and 12 European sites (Manninen et al., 2010). Such size dependency implies the involvement of different condensing vapors in the growth of different-size particles based on their saturation vapor pressures. The difference in GR between negative and positive polarities was minimal at both sites. The median GR of ions from 3 to 7 nm showed little difference between two sites. However, at SORPES, the median GR of ions from 7 to 20 nm ($GR^-$ 7-20 nm: 6.74 nm h$^{-1}$, $GR^+$ 7-20 nm: 6.82 nm h$^{-1}$) were higher than that at SMEAR II ($GR^-$ 7-20 nm: 5.58 nm h$^{-1}$, $GR^+$ 7-20 nm: 5.18 nm h$^{-1}$). This observation is consistent with the study by Manninen et al. (2010) on the GR of ions at various European sites, where they reported higher GR values at urban areas compared to rural and coastal sites. Similarly, in Finland, the GR of ions in the urban Helsinki was found to be higher than that in the clean Hyytiälä (Hussein et al., 2008). The higher GR at SORPES could be caused by more abundant condensing vapors that facilitate the growth process (Qi et al., 2018).

**Table 1**: Formation rates of 2 nm ions for both negative and positive polarities ($J_2^\pm$), formation rates of 2.5 nm total particles ($J_{2.5}^{total}$; sum of charged and neutral particles), growth rates of 3–7 nm and 7-20 nm ions for both negative and positive polarity ($GR^+$ and $GR^-$), and ion-induced fraction (the ratio of 2 nm ions and 2.5 nm total formation rates) at SMEAR II and SORPES. Formation rate of ions/particles and ion-induced fraction were determined during each NPF event from 9:00-15:00 LT at both sites.

| | SMEAR II | | | | SORPES | | | |
|---|---|---|---|---|---|---|---|---|
| | Average | Median | 25[th] | 75[th] | Average | Median | 25[th] | 75[th] |
| $J_2^-$ (cm$^{-3}$ s$^{-1}$) | 0.060 | 0.033 | 0.014 | 0.071 | 0.070 | 0.028 | 0.015 | 0.053 |
| $J_2^+$ (cm$^{-3}$ s$^{-1}$) | 0.066 | 0.041 | 0.020 | 0.082 | 0.056 | 0.025 | 0.014 | 0.046 |
| $J_{2.5}^{total}$ (cm$^{-3}$ s$^{-1}$) | 0.822 | 0.348 | 0.161 | 0.726 | 2.672 | 4.171 | 1.677 | 12.265 |
| $GR^-$ 3-7 nm (nm h$^{-1}$) | 4.878 | 3.492 | 1.959 | 5.817 | 4.144 | 3.177 | 2.221 | 5.757 |
| $GR^+$ 3-7 nm (nm h$^{-1}$) | 4.311 | 3.157 | 1.723 | 5.896 | 10.595 | 4.272 | 2.514 | 7.000 |
| $GR^-$ 7-20 nm (nm h$^{-1}$) | 7.159 | 5.572 | 3.839 | 8.342 | 7.870 | 6.736 | 5.107 | 9.353 |
| $GR^+$ 7-20 nm (nm h$^{-1}$) | 8.469 | 5.179 | 3.623 | 8.086 | 7.495 | 6.818 | 5.099 | 9.305 |
| Ion-induced fraction | 0.234 | 0.199 | 0.158 | 0.284 | 0.022 | 0.013 | 0.007 | 0.023 |

### 3.3.3 The role of ions in new particle formation

To shed more light on the role of ions in the NPF process, we compared the ion-induced fraction (the ratio of charged 2 nm to total 2.5 nm particle formation rates) at both SMEAR II and SORPES. As shown in Table 1, the median ion-induced fraction at SMEAR II was 19.9% (mean 23.4%), well within the typical range of 1 to 30% observed at European sites (Manninen et al., 2010). However, at SORPES, the median induced fraction was only 1.3% (mean 2.2%), approximately fifteen times lower than that at SMEAR II. The higher ion-induced fraction at SMEAR II (median $J_{2.5}^{total}$: 0.35 cm$^{-3}$ s$^{-1}$) compared with SORPES (median $J_{2.5}^{total}$: 4.17 cm$^{-3}$ s$^{-1}$) is in accordance with previous findings that the contribution of ion-induced nucleation to total nucleation increases with a decreasing total formation rate (Manninen et al., 2010). It is noteworthy that the 1.3% ion-induced fraction at SORPES is considerably higher than the value reported by Herrmann et al. (2014) in 2014 for the same site (median 0.2%; mean 0.2%). This difference may be attributed to anthropogenic emission reductions since 2013, driven by extensive air quality control efforts in China. As a result of these reductions, total particle formation rates in the YRD region was observed to decline over the period 2013–2019 (Shen et al., 2022). Given that variations in the ion-induced fraction are primarily influenced by the formation rate of total particles, the ion-induced fraction may increase as air pollution improves in urban areas. Nevertheless, while this is a plausible explanation, it needs to be kept in mind that Herrmann et al. (2014) estimated the formation rate of 2 nm particles based on observed values of formation rate of 6 nm particles, which may cause uncertainty.

In line with most of the observations within polluted boundary layers (Hirsikko et al., 2011), our work suggests that the contribution of ion-induced nucleation to NPF in polluted areas is minor. Nevertheless, the role of ions in NPF cannot be overlooked. As shown in Fig. 10, the formation of charged particles starts around an hour earlier than neutral particle formation at SORPES, suggesting that the ion-induced nucleation could precede neutral nucleation in this polluted area. This phenomenon may be explained by the fact that small ions are more easily activated to grow than neutral particles under lower vapor supersaturations (Winkler et al., 2008), and that ion-induced nucleation pathways are more important or even dominant at low vapor concentrations (Wagner et al., 2017). As ion-induced nucleation may require lower precursor vapor concentrations than neutral pathways, charged particles may be more readily activated in a polluted urban environment with limited low volatile or condensable vapors. The earlier onset of ion-induced nucleation is linked to the relatively low condensable vapor concentrations in the morning, which enable charged clusters to activate earlier than neutral particles as vapor levels start to rise with the increase in solar radiation. Such early occurrence of ion-induced nucleation has also been observed in other field measurements in Europe (Manninen et al., 2010; Gonser et al., 2014).

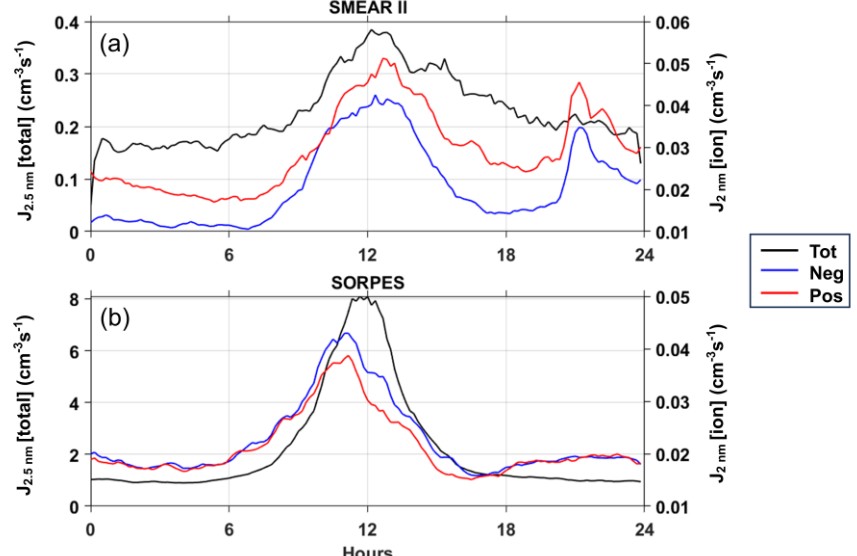

**Figure 10**: Diurnal variations in median charged and total particle formation rates during NPF event days at SMEAR II and SOREPS from June 2019 to August 2020. Charged (negative ions: blue line; positive ions: red line) and total formation rates (black line) are measured by NAIS in ion mode and particle mode, respectively.

In addition, the relationships between ion-induced fraction and the NPF ranking at SMEAR II and SORPES are presented in Fig. 11. The NPF ranking values were derived using the Nano Ranking Analysis method described by Aliaga et al. (2023). The NPF ranking values provide the information about the intensity of NPF events: on average, the days with higher (lower) ranking values have higher (lower) probability and intensity of NPF events (see Fig. S5 and S6). At SMEAR II, both ion formation rates and total particle formation rates increased with rising NPF ranking values (Fig.11a and 11c), while the ion-

induced fraction showed minimal change, with only a slight increase when ranking values were higher than 80% (Fig.11e). In contrast, at SORPES, ion formation rates showed little increase with rising ranking values, but particle formation rates

increased by orders of magnitude (Fig.11b and 11d). This significant difference led to a clear increase in the ion-induced fraction during periods of low NPF ranking values (<50%) (Fig.11f). At SORPES, the ion-induced fraction was highest when NPF ranking values were lowest (median of 3.2%, reaching up to 10.7%), which was up to 3 times higher than during periods with the highest NPF ranking values. The days with low ranking values are associated with the so-called "quiet" NPF (traditionally overlooked non-event days) as described by Kulmala et al. (2022b). Previous studies show that "quiet" NPF is a

non-negligible source of particles, especially in polluted environments (Kulmala et al., 2022b; Chen et al., 2023). Therefore, the relatively high ion-induced fraction observed during low ranking periods at SORPES suggests that ion-induced NPF still plays a notable role in airborne aerosol production in polluted urban environments. Furthermore, this phenomenon is consistent with the view of Kulmala et al. (2022b) that focusing on ion-induced NPF should be a priority in exploring the mechanisms of quiet NPF.

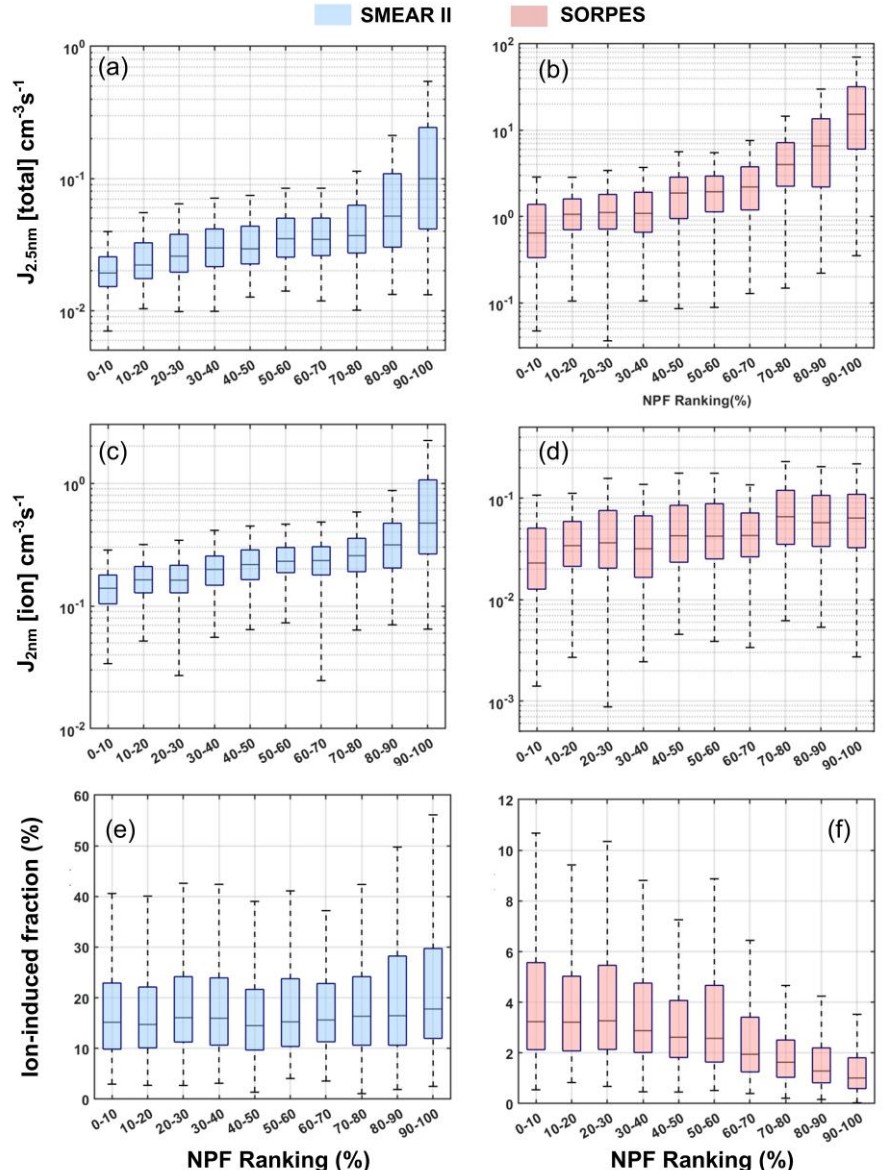

**Figure 11.** The total formation rate of 2 nm ions (sum of both polarities), formation rate of 2.5 nm total particles, ion-induced fraction (the ratio between charged 2 nm particles and 2.5 nm total particle formation rates) as a function of NPF ranking values at SMEAR II and SORPES during active time (9:00-15:00 LT) of NPF event days. The line inside each box is the median; the top and bottom of each box are the 25th and 75th percentiles, respectively; the whiskers are equal to 1.5 interquartile range.

## 4 Summary and conclusions

We conducted a comparative analysis of air ions in three size ranges (0.8-2 nm cluster ions; 2-7 nm intermediate ions, 7-20 nm large ions) at two "flagship" stations in the PEEX region: the SMEAR II site in a Finnish boreal forest and the SORPES site in an eastern Chinese megacity, covering the period from June 2019 to August 2020. At both sites, the differences in concentrations between the positive and negative cluster ions diminished with rising temperatures. This is likely due to the enhanced convective motions and turbulent mixing at higher air temperatures, which mitigate the Earth's electrode effect. During the whole measurement period, the median cluster ion concentration at SMEAR II (1270 cm$^{-3}$) was about six times greater than that at SORPES (220 cm$^{-3}$), which was caused by the high background aerosol loading in the urban area. Intermediate ion concentrations at two sites were very low and comparable, with median values of 27 cm$^{-3}$ and 25 cm$^{-3}$, respectively. The median large ion concentration at SORPES (197 cm$^{-3}$) was nearly three times higher than at SMEAR II (67 cm$^{-3}$). This difference is attributed to high probability of cluster ions coagulating with neutral particles at the polluted SORPES, with heavy traffic emissions further promoting this process.

Distinct seasonal and diurnal variations in ion concentrations were observed at both SMEAR II and SORPES. At both sites, cluster ion concentrations increased during summer, peaking in August at SMEAR II and in October at SORPES. Intermediate ion concentrations peaked during spring at SMEAR II and during autumn at SORPES, which was related to the higher frequency of NPF events in these seasons at the respective sites. Notably, a clear increase in negative intermediate ion concentration was observed at SORPES, which could be caused by the heavy and intensive rainfall in the YRD of eastern China. The diurnal cycles of ion concentrations in all the three size ranges were more pronounced during the warm season at both sites. At SORPES, ion concentrations generally peaked during daytime in the warm season. In contrast, at SMEAR II, ion concentrations increased during the darker hours of the warm season. It is noteworthy that the increase in nighttime cluster ion concentration at SMEAR II occurred throughout the warm part of the year (from April to September), while the nocturnal rise in intermediate ions was primarily observed in spring (from March to May).

A negative association was observed between the cluster ion concentration and CS at both sites. The negative correlation was stronger at SORPES compared to SMEAR II, suggesting that CS may be a decisive factor affecting cluster ion concentration in polluted urban area. Wind speed and direction also had a significant impact on ion concentrations at both sites. At SMEAR II, cluster ion concentration significantly decreases with an increasing wind speed. Additionally, ion concentrations were higher during northwesterly winds, correlating with NPF events that are favoured by clean winds originating from the pristine northern seas. At SORPES, intermediate and large ion concentrations were elevated with westerly winds, probably due to traffic emissions from the main roads to the southwest of the site.

We reported ion formation rates and growth rates (GR) for both polarities at the two sites. During the active time (9:00-15:00 LT) of NPF event days, the median ion formation rates, $J_2^{\mp}$ and $J_3^{\mp}$, at SMEAR II ($J_2^-$: 0.033 cm$^{-3}$ s$^{-1}$, $J_2^+$: 0.041 cm$^{-3}$ s$^{-1}$; $J_3^-$: 0.012 cm$^{-3}$ s$^{-1}$, $J_3^+$: 0.016 cm$^{-3}$ s$^{-1}$) were similar to those at SORPES ($J_2^-$: 0.028 cm$^{-3}$ s$^{-1}$, $J_2^+$: 0.025 cm$^{-3}$ s$^{-1}$; $J_3^-$: 0.028 cm$^{-3}$ s$^{-1}$, $J_3^+$: 0.027 cm$^{-3}$ s$^{-1}$). During the entire measurement period, $J_2^{\mp}$ and $J_3^{\mp}$ were higher during the warmer part of the year at both

sites. The median GR of ions from 3 to 7 nm showed a minimal difference between SMEAR II ($GR^-$ 3-7 nm: 3.50 nm h$^{-1}$, $GR^+$ 3-7 nm: 3.16 nm h$^{-1}$) and SORPES ($GR^-$ 3-7 nm: 3.18 nm h$^{-1}$, $GR^+$ 3-7 nm: 4.28 nm h$^{-1}$). At SORPES, however, the median GR of ions from 7 to 20 nm at SORPES ($GR^-$ 7-20 nm: 6.74 nm h$^{-1}$, $GR^+$ 7-20 nm: 6.82 nm h$^{-1}$) were higher than that at SMEAR II ($GR^-$ 7-20 nm: 5.58 nm h$^{-1}$, $GR^+$ 7-20 nm: 5.18 nm h$^{-1}$). Nighttime increases in $J_2^{\mp}$ during the NPF event days

were noted at SMEAR II. The median ion-induced fractions were estimated to be 19.9% at SMEAR II and 1.3% at SORPES, suggesting that the contribution of ions to NPF is minor in polluted environment. Nevertheless, we found that charged particles were activated earlier than neutral particles at SORPES, suggesting charged particles were more readily activated than neutral particles in polluted urban environment. High ion-induced fraction was observed during low NPF ranking periods at SORPES. Such days with low NPF intensity refer to "quiet" NPF, which is a significant source of aerosols in urban areas. The higher

ion-induced fraction observed during the typically overlooked quiet NPF compared to strong NPF at SORPES indicates that ion-induced NPF still plays a notable role in airborne aerosol production in polluted urban environments. We therefore highlight the need for long-term observations of air ions in both pristine boreal forests and polluted urban environments.

**Data availability**

The meteorological data from the SMEAR II can be accessed from the smartSMEAR website: http://avaa.tdata.fi/web/smart/. The ion and particle data from SMEAR II, as well as the measurement data from SORPES, are available from the corresponding authors upon request.

**Author contributions**

MK and XQ conceptualized the original idea. TZ performed the data analysis and wrote the paper under the supervision of XQ and JL. JL, LC, and XC were responsible for ion measurements at SMEAR II and SORPES. All authors contributed to the discussion of the results and provided input for the paper.

**Competing interests**

At least one of the (co-)authors is a member of the editorial board of Atmospheric Chemistry and Physics, and the authors also have no other competing interests to declare.

**Acknowledgments**

Xiang Li, Lian Duan, the SMEAR II and SORPES technical and scientific staff, and everyone else contributing to the
590 measurements are gratefully acknowledged.

**Financial support**

This work has been supported by the ACCC Flagship funded by the Academy of Finland (grant nos. 337549), Academy of Finland project (grant nos. 316114, 325647, 325681 and 347782), the "Gigacity" project funded by the Jenny and Antti Wihuri Foundation, and the National Natural Science Foundation of China (grant nos. 42175113).

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
