# Peer review of "Differential characterization of air ions in boreal forest of Finland and megacity of eastern China"

_EGUsphere, 2024_

## Author Response (AR1)

**Author's responses**

**Reply on RC1**

**General comments:**

Zhang et al. present a comprehensive analysis of long-term in-situ air ion measurements in an urban environment, with comparisons to data from a boreal forest site. The study thoroughly investigates seasonal and diurnal trends and includes calculations of particle formation and growth rates. The results are consistent with existing literature and reflect the expected behavior of air ions in these contrasting environments. While the work is detailed and well-executed, it primarily reinforces previously established knowledge. One of the key findings is the observed decline in ion concentrations at the SORPES station compared to earlier measurements; however, the authors note that this decrease may be influenced by differences in instrumentation. Overall, the manuscript is best considered a detailed measurement report rather than a source of novel conceptual insights. I recommend the manuscript for acceptance after the authors address the following minor comments:

**Response:** We sincerely thank the referee for the constructive comments, which have greatly contributed to improving our manuscript. This study presents the first long-term characterization of the formation and growth rates of charged particles in the western Yangtze River Delta region. By comparing the long-term dataset from two different sites, we demonstrate that ion-induced nucleation can precede neutral nucleation and play a non-negligible role in aerosol production in urban environments. The changes in the revised manuscript are highlighted in yellow. Below are our point-by-point responses to each comment.

**Specific comments:**

**Comment 1, Page 1 line 22:** The reader might not directly understand what is meant by the "electrode effect". Consider replacing by "Earth's electrode effect" or "fair-weather electric field near the ground"

**Response:** We thank the referee for pointing out that the term "electrode effect" may be ambiguous. In response, we have replaced "electrode effect" with "Earth's electrode effect" on the Page 1, Line 20.

**Comment 2, Page 1 line 28:** The authors report here ion formation rates at 3 nm which are otherwise not shown or referred to throughout the manuscript. Why did the authors chose to report ion formation rates at 2 nm and total formation rates at 2.5 nm in the main text? Why they did not report the ion and particle formation rates at the same size to make the calculation of the ion-induced fraction more reasonable?

**Response:** We thank the referee for highlighting that ion formation rates at 3 nm are not central to our analysis. Accordingly, we have removed the report of 3 nm ion formation rates in the abstract.

In the main text, we chose to calculate the ion formation rate at 2 nm and the total particle formation rate at 2.5 nm for the following reasons:

1. This choice allows us to only focus on the contribution of ion-induced nucleation: assessing the ion-induced fraction of NPF beyond 2 nm would include neutralization of charged clusters by ion-ion recombination and thus conflate ion-induced and ion-mediated fraction.

2. We calculate the total particle formation rate at 2.5 nm to avoid contamination from the overlapping corona-charger ions of NAIS in total particle measurements (Asmi et al., 2009; Manninen et al., 2011). Theoretically, the formation rate at 2 nm would be somewhat higher than that at 2.5 nm , which might lead to slight overestimation of the derived ion-induced fraction at 2 nm. We have explicitly acknowledged this potential uncertainty in the revised manuscript (Page 7, Line 190-193).
   "*In previous studies, the total particle formation rate was typically taken at 2 nm. To avoid contamination from charger ions in the NAIS total particle measurements, the total particle formation rate was calculated at 2.5 nm in this study.* **This methodological difference might introduce a slight overestimation, since the formation rate at 2.5 nm is theoretically lower than that at 2 nm**"

**Comment 3, Page 1 line 25:** The sentence needs rephrasing. Consider: "likely due to the efficient scavenging of cluster ions by the high number density of neutral aerosol particles prevalent in a megacity environment"

**Response:** We thank the referee for the helpful comment. We have rephrased the sentence on Page 1, Line 23-24 accordingly.
"*The median large ion concentration at SORPES was nearly three times higher (197 cm−3) than that at SMEAR II (67 cm−3),* **which is due to the high number density of neutral aerosol particles facilitating ion attachment in the polluted megacity environment**."

**Comment 4, Page 1 line 27:** Decisive factor for what? Be clearer: for reducing the cluster ion concentrations.

**Response:** We thank the referee for noting the ambiguity. We have revised the phrase on Page 1, Line 25-26 to read.

**Comment 5, Page 2 line 43:** 'their role' instead of 'their roles'

**Response:** We thank the referee for pointing out this wording issue. We have replaced "their roles" with "their role" on Page 2, Line 43.

**Comment 6, Page 2 line 55-58:** Some references cited here pertain to ion-mediated nucleation, while others pertain to ion-induced nucleation. It is important to recognize that these two terms

describe different pathways

**Response:** We thank the referee for highlighting the difference between ion-induced nucleation (IIN) and ion-mediated nucleation (IMN). IIN refers to the formation of new particles from small ionic clusters, preserving the ion charge during growth. In contrast, IMN includes IIN but also consider the interactions of ions and particles, such as recombination and attachment. To avoid confusion and correctly cite the references, we have updated the sentence on Page 2, Lines 54–57. The revised sentence now reads:

*"Besides the chamber experiments, model results indicate that ion-mediated nucleation (which includes ion-induced nucleation but also takes into account interactions between ions and particles, e.g., recombination and attachment) could be the dominant pathway for NPF (Yu and Turco, 2011; Yu, 2010)."*

**Comment 7, Page 3 line 86:** Why the Wang et al. 2020, reference has the author's first name?

**Response:** We thank the referee for noting this formatting inconsistency. We have corrected the reference in accordance with the journal's style (Page 3, Line 87).

**Comment 8, Page 4 Figure 1:** The authors should include credits or copyright statement for the map.

**Response:** We thank the reviewer for pointing this out. We have added the appropriate credit in the caption of Figure 1. The revised caption now reads (Page 4, Figure 1):
*Figure 1. Locations of SMEAR II in Hyytiälä, Finland and SORPES in Nanjing, eastern China on the map (from google earth) with population density.*

**Comment 9, Page 4 line 108:** SMEAR II **is** situated

**Response:** We thank the reviewer for pointing out the incomplete phrasing. We have revised the sentence to read. This change has been made on Page 4, Line 109.

**Comment 10, Page 5 line 133:** The NAIS data inversion is performed using the Spectops software and is not described by Wagner et al. The authors appear to conflate data inversion with the correction introduced by Wagner et al. Specifically, the Wagner et al. paper does not present inversion kernels; instead, it proposes a correction function to account for the detection efficiency of ions by the NAIS. Therefore, it cannot be claimed by the authors that the NAIS inversion kernel is calibrated based on Wagner et al., 2016 method.

**Response:** We thank the referee for pointing out this error. Indeed, the NAIS data inversion is not based on Wagner et al. (2016), but follows the procedure described by Manninen et al. (2016). We have revised the text on Page 5, Line 134-135 to read:
*"The NAIS data inversion was performed using the Spectops software following the procedure described by Manninen et al. (2016)".*

**Comment 11, Page 5 line 134:** This statement is also incorrect. The inversion performed by the Spectops software already accounts for internal losses within the NAIS. Corrections for diffusion losses in the sampling tube are typically applied after the data inversion.

**Response:** Thank you for your comment. We have revised the statement on Page 5, Line 135-136. It now reads:
*"Diffusion losses in the sampling tube were corrected after the data inversion."*

**Comment 12, Page 5 line 136:** The authors say here that the NAIS was unavailable from Sep 14 to October 15th. It is important to include this information in the captions of figure 3 and 4 as well.

**Response:** We thank the referee for this suggestion. We have added the following sentence to the captions of Figures 3 and 4:
*"Note that NAIS data were unavailable at SORPES from 14 September 2019 to 15 October 2019 due to instrument deployment in a short-term campaign."*

**Comment 13, Page 5 line 139:** Are the authors certain that a twin DMPS system is used at the SORPES station? A measurement range of 6 to 800 nm seems unusual for a typical twin DMPS setup. This range is more typical for the Helsinki custom-made flow switching DMA.

**Response:** We thank the referee for this comment. After verification, we confirm that during the measurement period (7 June 2019–31 August 2020), the SORPES site did **not** use the flow switching DMPS. Instead, the particle number size distribution from 4 to 500 nm was measured by two Scanning Mobility Particle Sizers (SMPS, TSI Inc.), i.e. nano-SMPS and long-SMPS, which consisted of different differential mobility analyzer (DMA) and condensation particle counter (CPC). The particle number size distribution from 500 nm to 20µm (aerodynamic diameter) was measured by an Aerodynamic Particle Sizer (APS, TSI Inc.). The SMPS data were combined with the APS data according to the method described by Beddows et al. (2010) in order to obtain the particle number size distribution from 4 nm to 1000 nm used in this study. We have modified the corresponding text (Page 5, Lines 138–145) to read:
*"At SMEAR II, particle number size distributions ranging from 3 to 1000 nm were additionally measured using a twin differential mobility particle sizer (DMPS). The twin DMPS measurement system contains two setups, each comprising a cylindrical DMA and a condensation particle counters (CPC). Detailed description for DMPS instrumentation can be found from Aalto et al. (2001). At SORPES, particle size distribution from 4 to 500 nm was measured by two Scanning Mobility Particle Sizers (SMPS, TSI Inc.), i.e. nano-SMPS and long-SMPS, which consisted of different DMA and CPC. Concurrently, an Aerodynamic Particle Sizer (APS, TSI Inc.) covered the size range from 500 nm to 20µm (aerodynamic diameter). The SMPS data were combined with the APS data according to the method described by Beddows et al. (2010) to obtain the particle number size distribution from 4 nm to 1000 nm used in this study."*

The summary of the instrumentation in Table SI has also been revised.

**Comment 14, Page 7 line 187:** Reference here should be to equations 3 and 4, not 4 and 5

**Response:** We thank the referee for catching this error. We have corrected the equation references on Page 7, Line 190, so that they now read "Eqs. 3 and 4" instead of "Eqs. 4 and 5."

**Comment 15, Page 7 line 208-209:** "As cluster ions are continually produced in the atmosphere and have shorter lifetimes compared to larger ions, they have a higher probability of attaching to neutral particles"
The sentence needs to be rephrased. The short lifetime of the small ions is due to their higher coagulation probability with bigger neutral particles and not vice versa.

**Response:** We thank the referee for this comment. We have rephrased the sentence on page 8, line 214-215 to ensure logical consistency with the surrounding context. The revised sentence now reads:
*"Since cluster ions are continually produced in the atmosphere and have a high coagulation probability with bigger neutral particles in the air, this attachment process may serve as a substantial source of large ions."*

**Comment 16, Page 8 line 211-212:** Please also include the recent study published in PNAS on braking as a source of highly charged aerosol particles (https://doi.org/10.1073/pnas.2313897121) to the list of references about ions and traffic emissions.

**Response:** We thank the reviewer for bringing this recent study to our attention. We have added this publication to the list of references about ions and traffic emissions on Page 8, Line 218

**Comment 17, Page 8 line 219-220:** "This phenomenon could be caused by a fraction of very small negative ions being cut out due to the lower limit of the NAIS (Figure 2a-b)".
If this is the case, then something similar should also be seen at the SMEAR II station. The distribution of negative ions there—although having a slightly higher peak diameter—is also not completely measured by the NAIS.

**Response:** We thank the referee for this insightful comment. Indeed, since the NAIS lower-size cutoff affects both sites, the absence of a similar feature at SMEAR II indicates that truncation by the instrument cannot explain the different polarity at two sites. Accordingly, we have removed the sentence attributing the phenomenon to NAIS detection limits (Page 8, Lines 219–220). Instead, as suggested in our response to Comment 18, we now introduce the proton-affinity argument first to explain the higher positive cluster ion concentrations at SORPES.

**Comment 18, Page 8 lines 223-225:** "Additionally, the elevated positive cluster ion concentration

at SORPES may indicate that SORPES hosts more air ions with the compounds containing the highest proton affinities, allowing them to capture positive charges."

As this is the most likely explanation for the difference between the two sites, please consider presenting this argument first.

**Response:** We thank the referee for this suggestion. We have restructured the text on Page 8, Lines 225–229 to present this argument first. The revised paragraph now reads:

*"In contrast, at SORPES, the concentration of positive cluster ions was higher than that of negative cluster ions.* ***This phenomenon may be attributable to SORPES hosting more air ions with the compounds containing the highest proton affinities, allowing them to capture positive charges.*** *The higher concentration of positive cluster ions at SORPES may also be partly due to the electrode effect of the negatively charged Earth's surface, which typically attracts small positive ions downward, causing a dominance of positive polarity near the ground in calm air (Hoppel, 1967; Wilson, 1924)"*

**Comment 19, Page 8 lines 227-228**: "As the mean size increases with air temperature, fewer negative ions fall below 0.8 nm, leading to a reduction in the concentration difference."

This hypothesis can be easily tested by plotting the size distribution at different temperatures. Please consider adding such a plot to the supplement.

**Response:** Fig. R1a–b confirms our response to Comment 17 that the NAIS lower-size cutoff affects the negative cluster ion concentration at both SMEAR II and SORPES. However, Fig. R1a–b does not support the assumption that fewer negative ions remain below 0.8 nm with increasing temperature. Consequently, this hypothesis cannot be confirmed. Therefore, we have removed this hypothesis from the manuscript (Page 8, Lines 227–228).

[Figure]

**Fig. R1:** Median ion size distribution of both negative and positive polarities from 0. 8-4 nm at SMEAR II (a,c) and SORPES (b,d) across different temperature intervals over the entire measurement period.

**Comment 20, Pages 9-10 lines 246-247:** "A possible explanation could be the elimination of large radiation sources near SORPES, particularly those from industrial activity".
Please give a reference to this statement or explain more.

**Response:** We thank the referee for this comment. Previous studies have shown that industrial ionizing radiation sources, such as nuclear facilities (Hörrak et al., 1994; Israelsson and Knudsen, 1986) or power lines (Matthews et al., 2010), could be the source of cluster ions. However, there is no published documentation on the closure of nearby industrial radiation sources during our study period substantiates this claim. To avoid the unsupported speculation, we have removed the sentence from Pages 9-10, Lines 246-247. The primary reason of the decrease in cluster ion concentration at SORPES could be the rapid urbanization in Nanjing over the past decade, including the transformation of unpaved roads to cemented surfaces, which has likely reduced radon emissions from the ground (Page 9-10, 249-251).

**Comment 21, Page 12 line 308:** Rephrase sentence. The concentration of cluster ions would be zero if the sink is higher than the production

**Response:** We thank the referee for this comment. We have rephrased the sentence on Page 12, line 309-311 to read:
*"The absence of nighttime, and even daytime, bursts in cluster ion concentration at SORPES may be attributed to high background particle loading, which likely keeps the sink of cluster ions consistently high throughout the day."*

**Comment 22, Page 14 figure 5 and associated discussion:** While the CS calculation method is consistent across both sites, the particle size distribution measured at SORPES covers a narrower size range compared to that at SMEAR. It is important that the same size range is considered in the CS calculation at both sites to ensure comparability. That said, the SORPES site is likely influenced by larger particles outside the detection range of standard DMPS systems, which could lead to an underestimation of the CS at this location. The authors are encouraged to include measurements of bigger particles from instruments such as OPCs or APS, in the calculation of CS if these are available.

**Response:** We thank the referee for raising this important point. We confirmed that during the study period, particle number size distribution in the size range from 4 nm to 1000 nm was observed by two sets of SMPS together with APS (see our response to Comment 13) at SORPES. At SMEAR II, particle number size distribution in the size range from 3 nm to 1000 nm was observed by a twin DMPS. Therefore, the size ranges used for CS calculations are effectively the same at SMEAR II (3-1000 nm)

and SORPES (4-1000 nm), ensuring direct comparability of CS at both sites. We have clarified it in the revised manuscript.

**Comment 23, Page 17 Lines 414- 416:** "The peak in large ion concentration at SORPES (13:00 LT) appeared earlier than at SMEAR II (18:00 LT), which was partly caused by a higher growth rate of newly formed particles at SORPES".

It is also worth mentioning here that the increase in intermediate ions also started earlier

**Response:** We thank the referee for this comment. We have updated the text on Page 17, Lines 414–415 to include the timing of the intermediate ion increase. The added sentence now reads:

*"The peak in intermediate ion concentration at SORPES (11-12:00 LT) appeared earlier than at SMEAR II (13:00 LT)."*

**Comment 24, Page 21 lines 485-493:** Please cite here the important work of Gonser et. al (2014) providing a mechanistic explanation for the time difference between ion and total formation rates (https://doi.org/10.5194/acp-14-10547-2014)

**Response:** We thank the referee for highlighting this key study. We have supplemented our discussion by citing this study on Page 21, Lines 497.

**Comment 25, Page 22 Line 504**: "with only a slight increase when ranking values were higher than 80% (Fig.11d)"

The reference here should be to figure 11e.

**Response:** Thanks for catching this error. We have replaced "Fig. 11d" with "Fig. 11e" on Page 22, Line 508.

**Comment 26, Page 22 Lines 505-506:** "at SORPES, ion formation rates showed little increase with rising ranking values, but particle formation rates increased by orders of magnitude (Fig.11b and 11d)."

The text is correct here, but the figures seem to have the wrong y-labels. Figure 11b should be the ion formation rate, while figure 11d should correspond to the particle formation rate as evidenced by the higher formation rates.

**Response:** We thank the referee for this comment. The positions for Figure 11b and Figure 11d were inadvertently swapped. We have updated Figure 11 on page 23 so that Figure 11b and Figure 11d are now correctly positioned and display the proper y-axis labels.

**Comment for Reference list:**

- Hirsikko, Yli-Juuti et al., 2007 : included twice in the reference list.
- Harrison & Carslaw, 2003: reference is not complete, needs page numbers or document

number

- Laakso et al., 2002: reference page number or document number is incorrect
- Iida et al., 2006: document number is not correct: J. Geophys. Res., 111, D23201
- Gagne et al., 2011: missing the journal information
- Leino et al., 2016: missing the journal information
- Hari et al., 2013: this is not the proper way to cite a chapter in a book
- Wagner et al., 2017: reference needs journal volume and page number
- Aliaga et al., 2023: must reference the final articles and not the discussion
- Komppula et., 2007: missing page numbers
- Mirme et., 2007: missing page numbers
- Hoppel 1986: could be a wrong reference as the mentioned publication have two other authors than Hoppel, R. V. Anderson, and John C. Willett. The right reference is Hoppel, W. A. (1967). Theory of the electrode effect, J. Atmos. Terrest. Phys. 29 , 709
- Nieminen et al., 2014: publication details missing
- Buenrostro et al., 2016: publication details missing

**Response:** We thank the referee for the comment. We have revised the referenced accordingly and updated the References section on Pages 26–36.

**Reference:**

Aalto, P., Hämeri, K., Becker, E., Weber, R., Salm, J., Mäkelä, J. M., Hoell, C., O'dowd, C. D., Hansson, H.-C., and Väkevä, M.: Physical characterization of aerosol particles during nucleation events, Tellus B, 53, 344-358, https://doi.org/10.3402/tellusb.v53i4.17127, 2001.

Asmi, E., Sipilä, M., Manninen, H., Vanhanen, J., Lehtipalo, K., Gagné, S., Neitola, K., Mirme, A., Mirme, S., and Tamm, E.: Results of the first air ion spectrometer calibration and intercomparison workshop, Atmos. Chem. Phys., 9, 141-154, https://doi.org/10.5194/acp-9-141-2009, 2009.

Beddows, D. C., Dall'Osto, M., and Harrison, R. M.: An enhanced procedure for the merging of atmospheric particle size distribution data measured using electrical mobility and time-of-flight analysers, Aerosol. Sci. Tech., 44, 930-938, https://doi.org/10.1080/02786826.2010.502159, 2010.

Hoppel, W.: Theory of the electrode effect, J. Atmos. Terr. Phys., 29, 709-721, 1967.

Hörrak, U., Iher, H., Luts, A., Salm, J., and Tammet, H.: Mobility spectrum of air ions at Tahkuse Observatory, J. Geophys. Res.-atmos., 99, 10,697-610,697, https://doi.org/10.1029/93JD02291, 1994.

Israelsson, S. and Knudsen, E.: Effects of radioactive fallout from a nuclear power plant accident on electrical parameters, J. Geophys. Res.-atmos., 91, 11909-11910, https://doi.org/10.1029/JD091iD11p11909, 1986.

Manninen, H. E., Mirme, S., Mirme, A., Petäjä, T., and Kulmala, M.: How to reliably detect molecular clusters and nucleation mode particles with Neutral cluster and Air Ion Spectrometer (NAIS), Atmos. Meas. Tech., 9, 3577-3605, https://doi.org/10.5194/amt-9-3577-2016, 2016.

Manninen, H. E., Franchin, A., Schobesberger, S., Hirsikko, A., Hakala, J., Skromulis, A., Kangasluoma, J., Ehn, M., Junninen, H., and Mirme, A.: Characterisation of corona-generated ions used in a Neutral cluster and Air Ion Spectrometer (NAIS), Atmos. Meas. Tech., 4, 2767-2776, https://doi.org/10.5194/amt-4-2767-2011, 2011.

Matthews, J. C., Ward, J. P., Keitch, P. A., and Henshaw, D. L.: Corona ion induced atmospheric potential

gradient perturbations near high voltage power lines, Atmos. Environ., 44, 5093-5100, https://doi.org/10.1016/j.atmosenv.2010.09.007, 2010.

Wagner, R., Manninen, H. E., Franchin, A., Lehtipalo, K., Mirme, S., Steiner, G., Petäjä, T., and Kulmala, M.: On the accuracy of ion measurements using a Neutral cluster and Air Ion Spectrometer, Boreal. Environ. Res., 21, 230-241, 2016.

Wilson, C. T. R.: The electric field of a thundercloud and some of its effects, P. Phys. Soc. Lond., 37, 32D, 1924.

Yu, F.: Ion-mediated nucleation in the atmosphere: Key controlling parameters, implications, and look-up table, J. Geophys. Res.-atmos., 115, https://doi.org/10.1029/2009JD012630, 2010.

Yu, F. and Turco, R.: The size-dependent charge fraction of sub-3-nm particles as a key diagnostic of competitive nucleation mechanisms under atmospheric conditions, Atmos. Chem. Phys., 11, 9451-9463, https://doi.org/10.5194/acp-11-9451-2011, 2011.

**Reply on RC2**

**General comments:**

This study reported differential characterization of air ions in the boreal forest of Finland and the megacity of eastern China. The authors presented some interesting results. The manuscript is well-written. After the following questions are considered, it would be worthy of publication in ACP.

**Response:** We would like to thank the referee for the insightful comments, which have greatly improved our manuscript. The changes in the revised manuscript are highlighted in yellow. Below are our point-by-point responses to each comment.

**Specific comments:**

**Comment 1:** Air temperature was identified as a crucial factor influencing ion polarity. Thus, the temperature of the instruments themselves might affect ion polarity. Consequently, this could introduce bias at the two sites. How can these biases be avoided?

**Response:** During measurement periods at SMEAR II and SOPRES, the NAIS instruments were operated according to the standard operating procedure provided by Manninen et al. (2016). At both sites, each NAIS was placed indoors in an air-conditioned environment maintained at approximately 25 °C, ensuring stable instrument temperatures and thus preventing potential temperature-induced biases or malfunctions. Therefore, the temperature of the instruments themselves is unlikely to affect the ion polarity measurements. As discussed in the manuscript on Page 8, line 226-231, we consider ambient air temperature to be the primary factor introducing differences in ion polarity at two sites. This is likely because elevated air temperatures enhance convective motions and turbulent mixing, thereby diminishing the Earth's electrode effect and influencing ion polarity.

To clarify that instrument temperature itself does not introduce such biases, we have

added the following statement to Section 2.2 (Instrumentation, Page 5, line 133-134): *"At both sites, each NAIS was placed indoors in an air-conditioned environment maintained at around 25 °C to ensure stable instrument temperatures and thus preventing potential temperature-induced biases or malfunctions."*

**Comment 2:** What are the reasons that ion-induced nucleation could precede neutral nucleation in this polluted environment?

**Response:** We thank the referee for the comment. As discussed on Page 21, line 489-497, we propose several possible reasons why ion-induced nucleation could precede neutral nucleation in a polluted environment:

1. Small ions are more likely to be activated to growth than neutral particles at lower vapor supersaturations and ion-induced formation pathways are more important or even dominating at low vapor concentrations (Winkler et al., 2008; Wagner et al., 2017)
2. The earlier onset of ion-induced nucleation may be associated with the relatively low vapor concentrations in the early morning, which facilitate the activation of charged clusters before neutral particles as vapor levels begin to rise with the increase in solar radiation.

To clarify these points, we have revised the text on Page 21, Lines 488–497 to read: *"As shown in Fig. 10, the formation of charged particles starts around an hour earlier than neutral particle formation at SORPES, suggesting that the ion-induced nucleation could precede neutral nucleation in this polluted area. This phenomenon may be explained by the fact that small ions are more easily activated to grow than neutral particles under lower vapor supersaturations (Winkler et al., 2008), and that ion-induced nucleation pathways are more important or even dominant at low vapor concentrations (Wagner et al., 2017). As ion-induced nucleation may require lower precursor vapor concentrations than neutral pathways, charged particles may be more readily activated in a polluted urban environment with limited low volatile or condensable vapors. The earlier onset of ion-induced nucleation may be linked to the relatively low condensable vapor concentrations in the morning, which enable charged clusters to activate earlier than neutral particles as vapor levels start to rise with the increase in solar radiation. Such early occurrence of ion-induced nucleation has also been observed in other field measurements in Europe (Manninen et al., 2010; Gonser et al., 2014)"*

**Comment 3, Lines 445–455:** During new particle formation (NPF) events, both charged and neutral particles are present in the same atmospheric environment. What is the difference in their growth processes?

**Response:** We thank the referee for this insightful question. Compared to neutral particles, charged clusters activate more readily and grow faster during the initial stages of

NPF (Winkler et al., 2008), consistent with our observation that ion-induced nucleation can precede neutral nucleation. This might be because the electric charges increase the condensation of polar vapors on the charged clusters due to increased collision rate and/or enhance their growth by making the forming clusters more stable (decreased evaporation rate) (Laakso et al., 2003; Lehtipalo et al., 2016). Once both ion and particle modes have formed during the growth process, the later-appearing neutral particles exhibit notably enhanced growth rates as particle size increases (Gonser et al., 2014). Consequently, the time difference between the earlier-occurring ion fraction and the total particle fraction gradually decreases, eventually vanishing completely. This apparent slowing of ion growth and acceleration of neutral particle growth indicate significant ion–particle interactions, including neutralization via ion–ion recombination and ion attachment to neutral clusters. These ion–particle interactions link and mutually influence the growth processes of charged and neutral particles, and this effect critically depends on the abundance of condensable vapors and the related strength of ion-induced nucleation (Gonzalez Carracedo et al., 2022)

We have modified the paragraph on Page 21, Lines 488–497 to discuss on the differences in the growth processes of charged and neutral particles.

**Comment 4, Lines 475–480:** As PM2.5 concentrations decreased in China, the atmospheric oxidation capacity increased. This could lead to an increase in H2SO4 concentrations. Particulate sulfate primarily forms in the particle phase. A reduction in PM2.5 does not necessarily indicate a decrease in gas-phase H2SO4 concentrations.

**Response:**  We thank the referee for the comment. Long-term observations in Beijing from 2013 to 2018 showed a decrease in particulate sulfate concentrations in submicrometer particles, whereas the concentration of gas-phase $H_2SO_4$ did not exhibit a clear declining trend during the same period (Li et al., 2020). Moreover, the long-term behavior of gas-phase $H_2SO_4$ in response to air quality control measures in China remains a subject of debate. Therefore, we acknowledge that there is insufficient evidence to conclude that a reduction in $PM_{2.5}$ necessarily indicates a decrease in gas-phase $H_2SO_4$ concentrations.

However, Shen et al. (2022) reported that in the Yangtze River Delta (YRD) region of China, total particle formation rates declined from 2013 to 2019 due to reductions in anthropogenic emissions. Accordingly, we have revised the discussion on Page 20, Lines 479–483 to state:

*"This difference may be attributed to anthropogenic emission reductions since 2013, driven by extensive air quality control efforts in China. As a result of these reductions, total particle formation rates in the YRD region was observed to decline over the period 2013–2019 (Shen et al., 2022). Given that variations in the ion-induced fraction are primarily influenced by the formation rate of total particles, the ion-induced fraction may increase as air pollution improves in urban areas."*

**Reference:**

Gonser, S. G., Klein, F., Birmili, W., Gröss, J., Kulmala, M., Manninen, H. E., Wiedensohler, A., and Held, A.: Ion–particle interactions during particle formation and growth at a coniferous forest site in central Europe, Atmos. Chem. Phys., 14, 10547-10563, https://doi.org/doi.org/10.5194/acp-14-10547-2014, 2014.

Gonzalez Carracedo, L., Lehtipalo, K., Ahonen, L. R., Sarnela, N., Holm, S., Kangasluoma, J., Kulmala, M., Winkler, P. M., and Stolzenburg, D.: On the relation between apparent ion and total particle growth rates in the boreal forest and related chamber experiments, Atmos. Chem. Phys., 22, 13153-13166, https://doi.org/10.5194/acp-22-13153-2022, 2022.

Laakso, L., Kulmala, M., and Lehtinen, K. E.: Effect of condensation rate enhancement factor on 3-nm (diameter) particle formation in binary ion-induced and homogeneous nucleation, J. Geophys. Res.-Atmos., 108, https://doi.org/10.1029/2003JD003432, 2003.

Lehtipalo, K., Rondo, L., Kontkanen, J., Schobesberger, S., Jokinen, T., Sarnela, N., Kürten, A., Ehrhart, S., Franchin, A., and Nieminen, T.: The effect of acid–base clustering and ions on the growth of atmospheric nano-particles, Nat. Commun., 7, 11594, https://doi.org/10.1038/ncomms11594, 2016.

Li, X., Zhao, B., Zhou, W., Shi, H., Yin, R., Cai, R., Yang, D., Dällenbach, K., Deng, C., and Fu, Y.: Responses of gaseous sulfuric acid and particulate sulfate to reduced SO2 concentration: A perspective from long-term measurements in Beijing, Sci. Total. Environ., 721, 137700, https://doi.org/10.1016/j.scitotenv.2020.137700, 2020.

Manninen, H. E., Mirme, S., Mirme, A., Petäjä, T., and Kulmala, M.: How to reliably detect molecular clusters and nucleation mode particles with Neutral cluster and Air Ion Spectrometer (NAIS), Atmos. Meas. Tech., 9, 3577-3605, https://doi.org/10.5194/amt-9-3577-2016, 2016.

Manninen, H. E., Nieminen, T., Asmi, E., Gagné, S., Häkkinen, S., Lehtipalo, K., Aalto, P., Vana, M., Mirme, A., and Mirme, S.: EUCAARI ion spectrometer measurements at 12 European sites–analysis of new particle formation events, Atmos. Chem. Phys., 10, 7907-7927, https://doi.org/doi.org/10.5194/acp-10-7907-2010, 2010.

Shen, X., Sun, J., Ma, Q., Zhang, Y., Zhong, J., Yue, Y., Xia, C., Hu, X., Zhang, S., and Zhang, X.: Long-term trend of new particle formation events in the Yangtze River Delta, China and its influencing factors: 7-year dataset analysis, Sci. Total. Environ., 807, 150783, https://doi.org/10.1016/j.scitotenv.2021.150783, 2022.

Wagner, R., Yan, C., Lehtipalo, K., Duplissy, J., Nieminen, T., Kangasluoma, J., Ahonen, L. R., Dada, L., Kontkanen, J., and Manninen, H. E.: The role of ions in new particle formation in the CLOUD chamber, Atmos. Chem. Phys., 17, 15181-15197, https://doi.org/10.5194/acp-17-15181-2017, 2017.

Winkler, P. M., Steiner, G., Vrtala, A., Vehkamäki, H., Noppel, M., Lehtinen, K. E., Reischl, G. P., Wagner, P. E., and Kulmala, M.: Heterogeneous nucleation experiments bridging the scale from molecular ion clusters to nanoparticles, Science, 319, 1374-1377, https://doi.org/10.1126/science.1149034, 2008.